# Understanding the Limitations of Deep Models for Molecular property prediction: Insights and Solutions

**Jun Xia,** **Lecheng Zhang,** **Xiao Zhu,** **Yue Liu, Zhangyang Gao,**
**Bozhen Hu, Cheng Tan, Jiangbin Zheng, Siyuan Li, Stan Z. Li**[†]
School of Engineering, Westlake University
{xiajun, zhanglecheng, stan.zq.li}@westlake.edu.cn

## Abstract

Molecular Property Prediction (MPP) is a crucial task in the AI-driven Drug Discovery (AIDD) pipeline, which has recently gained considerable attention thanks to advancements in deep learning. However, recent research has revealed that deep models struggle to beat traditional non-deep ones on MPP. In this study, we benchmark 12 representative models (3 non-deep models and 9 deep models) on 15 molecule datasets. Through the most comprehensive study to date, we make the following key observations: **(i)** Deep models are generally unable to outperform non-deep ones; **(ii)** The failure of deep models on MPP cannot be solely attributed to the small size of molecular datasets; **(iii)** In particular, some traditional models including XGB and RF that use molecular fingerprints as inputs tend to perform better than other competitors. Furthermore, we conduct extensive empirical investigations into the unique patterns of molecule data and inductive biases of various models underlying these phenomena. These findings stimulate us to develop a simple-yet-effective feature mapping method for molecule data prior to feeding them into deep models. Empirically, deep models equipped with this mapping method can beat non-deep ones in most MoleculeNet datasets. Notably, the effectiveness is further corroborated by extensive experiments on cutting-edge dataset related to COVID-19 and activity cliff datasets.

## 1 Introduction

Molecular Property Prediction (MPP) is a critical task in computational drug discovery, aimed at identifying molecules with desirable pharmacological and ADMET (absorption, distribution, metabolism, excretion, and toxicity) properties. Machine learning models have been widely used in this fast-growing field, with two types of models being commonly employed: traditional non-deep models and deep models. In non-deep models, molecules are fed into traditional machine learning models such as random forest and support vector machines in the format of computed or handcrafted molecular fingerprints [64]. The other group utilizes deep models to extract expressive representations for molecules in a data-driven manner. Specifically, the Multi-Layer Perceptron (MLP) could be applied to computed or handcrafted molecular fingerprints; Sequence-based neural architectures including Recurrent Neural Networks (RNNs) [43], 1D Convolutional Neural Networks (1D CNNs) [22], and Transformers [25, 54] are exploited to encode molecules represented in Simplified Molecular-Input Line-Entry System (SMILES) strings [71]. Additionally, molecules can be naturally represented as graphs with atoms as nodes and bonds as edges, inspiring a line of works to leverage such structured inductive bias for better molecular representations [20, 76, 79, 58]. The key advancements underneath these approaches are Graph Neural Networks (GNNs), which

---

[*]Equal Contribution.
[†]Corresponding author

37th Conference on Neural Information Processing Systems (NeurIPS 2023).

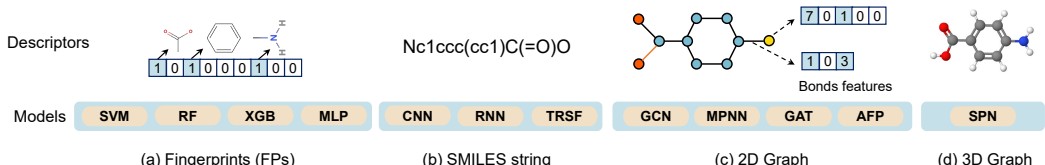

Figure 1: Exemplary molecular descriptors and their corresponding models in our benchmark. **SVM**: Support Vector Machine [84]; **RF**: Random Forest [61]; **XGB**: eXtreme Gradient Boosting [8]; **MLP**: Multi-Layer Perceptron; **CNN**: 1D Convolution Neural Network [32]; **RNN**: Recurrent Neural Network (GRU) [46]; **TRSF**: TRanSFormer [67]; **GCN**: Graph Convolution Network [33]; **MPNN**: Message-Passing Neural Network [20]; **GAT**: Graph Attention neTwork [68]; **AFP**: Attentive FP [77]; **SPN**: SPhereNet [38]. The above-mentioned abbreviations are applicable throughout the entire paper.

consider graph structures and attributive features simultaneously in the learning process [33, 68, 24]. More recently, researchers incorporate 3D conformations of molecules into their representations for better performance, whereas pragmatic considerations such as calculation cost, alignment invariance, uncertainty in conformation generation, and unavailable conformations of target molecules limited the practical applicability of these models [5, 17, 57, 16, 38]. We summarize the widely-used molecular descriptors and their corresponding models in our benchmark, as shown in Figure 1. Despite the fruitful progress, previous studies [41, 29, 79, 65, 30, 13, 66] have observed that deep models struggled to outperform non-deep ones on molecular datasets. However, these studies neither consider the emerging powerful deep models (e.g., Transformer [25], SphereNet [37]) nor explore various molecular descriptors (e.g., 3D molecular graph). Also, they did not investigate the reasons why deep models often fail on molecules.

To narrow this gap, we present the most comprehensive benchmark study on molecular property prediction to date, with a precise methodology for dataset inclusion and hyperparameter tuning. Our empirical results confirm the observations of previous studies, namely that deep models generally struggle to outperform traditional non-deep counterparts, even without accounting for the slower training of deep learning algorithms. Moreover, we observe several interesting phenomena that challenge the prevailing beliefs of the community, which can guide optimal methodology design for future studies.

Furthermore, we aim to understand why deep models often underperform non-deep ones in MPP. Specifically, we transform the original molecular data to observe the performance changes of various models, uncovering the unique patterns of molecular data and the differing inductive biases of various models. These in-depth empirical studies shed light on the benchmarking results: Deep models struggle to learn non-smooth target functions that map molecular data to labels, while the target functions are often non-smooth in MPP. This means that small changes in the chemical structure of a molecule may result in large changes in molecular properties. Additionally, deep models tend to attend to molecule features as a whole, especially handling the molecular fingerprints, while partial substructures known as functional groups are the most informative for molecules. On the other hand, XGB and random forest are well-suited for molecules because they make decisions based on each dimension of molecular features separately. Based on these phenomena and analyses, we develop a novel feature mapping method for molecule data before feeding them into models. Theoretically, we show that our method can help deep models learn non-smooth target functions that map molecules to properties. Moreover, our method is readily pluggable into various deep methods for performance improvement.

We highlight the following contributions: **(I)** We provide the most comprehensive benchmark on MPP tasks to date and expose the limitations of deep models on molecule datasets. Our findings offer new and valuable insights for the fast-growing AIDD community. **(II)** We empirically investigate the unique patterns of molecular data and inductive biases of various models, providing explanations for why deep models often cannot beat non-deep ones on MPP tasks. **(III)** We develop a simple-yet-effective feature mapping method to help deep models learn the non-smooth target functions with theoretical guarantees. **(IV)** We verify the effectiveness of our method through extensive experiments on MoleculeNet datasets, a cutting-edge dataset related to COVID-19 and activity cliff datasets.

## 2 Related Work

In this section, we elaborate on various molecular descriptors and their respective learning models.

### 2.1 Fingerprints-based Molecular Descriptors

Molecular fingerprints (FPs) serve as one of the most important descriptors for molecules. Typical examples include Extended-Connectivity Fingerprints (ECFP) [53] and PubChemFP [70]. These fingerprints encode the neighboring environments of heavy atoms in a molecule into a fixed bit string with a hash function, where each bit indicates whether a certain substructure is present in the molecule. Traditional models and MLPs can take these fingerprints as 'raw' input. However, the high-dimensional and sparse nature of FPs introduces additional efforts for feature selection when they are fed into certain models. Additionally, it is difficult to interpret the relationship between properties and structures because the hash functions are non-invertible.

### 2.2 Linear Notation-based Molecular Descriptors

Another option for molecules is linear notations, among which SMILES [71] is the most frequently-used one owing to its versatility and interpretability. In SMILES, each atom is represented as a respective ASCII symbol; Chemical bonds, branching, and stereochemistry are denoted by specific symbols. However, a significant fraction of SMILES strings does not correspond to chemically valid molecules. As a remedy, a new language named SELF-referencIng Embedded Strings (SELFIES) for molecules was introduced in 2020 [34]. Every SELFIES string corresponds to a valid molecule, and SELFIES can represent every molecule. Naturally, RNNs, 1D CNN, and Transformers are powerful deep models for processing such sequences [69, 86, 25, 55, 82]. However, the poor scalability of the sequential notations and the loss of spatial information limit the performances of these approaches.

### 2.3 2D and 3D Graph-based Molecular Descriptors

Molecules can be represented with graphs naturally, with nodes as atoms and edges as chemical bonds. Initially, [15] first adopted convolutional layers to encode molecular graphs to neural fingerprints. Following this work, [11] employs the atom-based message-passing scheme to learn expressive molecular graph representations. To complement the atom's information, [31] utilized both the atom's and bonds' attributes, and MPNN [20] generalized it to a unified framework. Also, multiple variants of the MPNN framework are developed to avoid unnecessary loops (DMPNN [79]), to strengthen the message interactions between nodes and edges (CMPNN [58]), to capture the complex inherent quantum interactions of molecules (MGCN [39]), or take the longer-range dependencies (Attentive FP [76]). More recently, some hybrid architectures [54, 81, 42, 45] of GNNs and transformers are emerging to capture the topological structures of molecular graphs.

The 3D molecular graph is composed of nodes (atoms), and their positions in 3D space and edges (bonds). The advantage of using 3D geometry is that the conformer information is critical to many molecular properties, especially quantum properties. In addition, it is also possible to directly leverage stereochemistry information such as chirality given the 3D geometries. Recently, multiple works [57, 56, 14, 38, 3] have developed message-passing mechanisms tailored for 3D geometries, which enable the learned molecular representations to follow certain physical symmetries, such as equivariance to translations and rotations. However, the calculation cost, alignment invariance, uncertainty in conformation generation, and unavailable conformations of target molecules limited the applicability of these models in practice.

## 3 Benchmark Representative Models on Multiple Molecular Datasets.

In this section, we present a benchmark on 15 molecular datasets with 12 representative models.

### 3.1 Experimental Setups

**Fingerprints $\longmapsto$ SVM, XGB, RF, and MLP.** Following the common practice [30, 63, 50], we feed the concatenation of various molecular fingerprints including 881 PubChem fingerprints (PubchemFP), 307 substructure fingerprints (SubFP), and 206 MOE 1-D and 2-D descriptors [80] to

SVM, XGB, RF, and MLP models to comprehensively represent molecular structures, with some pre-processing procedures to remove features (1) with missing values; (2) with extremely low variance (variance < 0.05); (3) have a high correlation (pearson correlation coefficient > 0.95) with another feature. The retained features are normalized to the mean value of 0 and variance of 1. Additionally, considering that traditional machine models (SVM, RF, XGB) cannot be directly applied in the multi-task molecular datasets, we split the multi-task dataset into multiple single-task datasets and use each of them to train the models. Finally, we report the average performance of these single tasks.

**SMILES strings ⟼ CNN, RNN, and TRSF.** We adopt the 1D CNNs from a recent study [32], which include a single 1D convolutional layer with a step size equal to 1, followed by a fully connected layer. As for the RNN, we use a 3-layer bidirectional gated recurrent units (GRUs) [10] with 256 hidden vector dimensions. Additionally, we use the pre-trained SMILES transformer [25] with 4 basic blocks and each block has 4-head attentions with 256 embedding dimensions and 2 linear layers. The SMILES are split into symbols (e.g., 'Br', 'C', '=', '(','2') and then fed into the transformer together with the positional encoding [67].

**2D Graphs ⟼ GCN, MPNN, GAT, and AFP.** As in previous studies [76], we exhaustively utilized all readily available atom/bond features in our 2D graph-based descriptors. Specifically, we have incorporated 9 atom features, including atom symbol, degree, and formal charge, using a one-hot encoding scheme. In addition, we included 4 bond features, such as type, conjugation, ring, and stereo. The resulting encoded graphs were then fed into GCN, MPNN, GAT, and AFP models. Further details on the graph descriptors used in our experiments can be found in [76].

**3D Graphs ⟼ SPN.** We employ the recently proposed SphereNet [38] for molecules with 3D geometry. Specifically, for quantum mechanics datasets (QM7, QM8, and QM9) that contain 3D atomic coordinates calculated with ab initio Density Functional Theory (DFT), we feed them into SphereNet directly. For other datasets without labeled conformations, we used RDKit [35]-generated conformations to satisfy the request of SphereNet.

**Datasets splits, evaluation protocols and metrics, hyper-parameters tuning.** Firstly, we randomly split the training, validation, and test sets at a ratio of 8:1:1. And then, we tune the hyper-parameters based on the performance of the validation set. Due to the heavy computational overhead, GNNs-based models on the HIV and MUV datasets are in 30 evaluations; all the models on the QM7 and QM8 are in 10 evaluations; all the models on the QM9 dataset are in one evaluation. And then, we conduct 50 independent runs with different random seeds for dataset splitting to obtain more reliable results, using the optimal hyper-parameters determined before. Following MoleculeNet benchmark [72], we evaluate the classification tasks using the area under the receiver operating characteristic curve (AUC-ROC), except the area under the precision curve (AUC-PRC) on MUV dataset due to its extreme biased data distribution. The performance on the regression task are reported using root mean square error (RMSE) or mean absolute error (MAE). Kindly note that we report the average performance across multi-tasks on some datasets because they contain more than one task. Additionally, to avoid the overfitting issue, all the deep models are trained with an early stopping scheme if no validation performance improvement is observed in successive 50 epochs. We set the maximal epoch as 300 and the batch-size as 128. *We provide more details including hyper-parameters tuning space for each model in the appendix.*

## 3.2 Observations

Table 1 documents the benchmark results for various models and datasets, from which we can make the following ***Observations***:

***Observation 1.*** **Deep models underperform non-deep counterparts in most cases.**
As can be observed in Table 1, non-deep models rank as the top one on 11/15 datasets. Kindly note that we report the results of each task in the QM9 dataset in the appendix. On some datasets such as MUV, QM7, and BACE, three non-deep models can even beat any deep models.

***Observation 2.*** **The failure of deep models should not be solely attributed to the limited size of molecular datasets.**
Intuitively, many previous works [21, 79, 59] pointed out that the small size of molecular datasets is a bottleneck for deep learning models and propose various strategies accordingly [49, 48]. Here, we complement this pre-dominant belief with a new opinion with empirical evidence. As shown in Table 1, all the non-deep models can outperform any deep ones on some larger-scale datasets (e.g.,

Table 1: The comparison of representative models on multiple molecular datasets. The standard deviations can be seen in the appendix for the limited space. **No.**: Number of the molecules in the datasets. The top-3 performances on each dataset are highlighted with the grey background. The best performance is highlighted with **bold**. Kindly note that **'TRSF'** denotes the transformer that has been pre-trained on 861, 000 molecular SMILES strings. The results on QM 9 can be seen in the appendix.

| Dataset (No.) | Metric | SVM | XGB | RF | CNN | RNN | TRSF | MLP | GCN | MPNN | GAT | AFP | SPN |
|---|---|---|---|---|---|---|---|---|---|---|---|---|---|
| BACE (1,513) | AUC_ROC | 0.886 | **0.896** | 0.890 | 0.815 | 0.559 | 0.835 | 0.887 | 0.880 | 0.846 | 0.886 | 0.879 | 0.882 |
| HIV (40,748) | AUC_ROC | 0.817 | 0.823 | 0.826 | 0.733 | 0.639 | 0.748 | 0.791 | **0.834** | 0.814 | 0.812 | 0.819 | 0.818 |
| BBBP (2,035) | AUC_ROC | 0.913 | **0.926** | 0.923 | 0.760 | 0.693 | 0.897 | 0.918 | 0.915 | 0.872 | 0.902 | 0.893 | 0.905 |
| ClinTox (1,475) | AUC_ROC | 0.879 | 0.919 | 0.933 | 0.685 | 0.813 | **0.963** | 0.890 | 0.889 | 0.868 | 0.891 | 0.907 | 0.912 |
| SIDER (1,366) | AUC_ROC | 0.626 | 0.638 | **0.644** | 0.591 | 0.515 | 0.641 | 0.617 | 0.633 | 0.603 | 0.614 | 0.620 | 0.613 |
| Tox21 (7,811) | AUC_ROC | 0.820 | 0.837 | 0.838 | 0.766 | 0.734 | 0.817 | 0.834 | 0.830 | 0.816 | 0.829 | **0.845** | 0.827 |
| ToxCast (8,539) | AUC_ROC | 0.725 | 0.785 | 0.778 | 0.735 | 0.74 | 0.780 | 0.781 | 0.767 | 0.736 | 0.768 | **0.788** | 0.772 |
| MUV (93,087) | AUC_PRC | **0.093** | 0.072 | 0.069 | 0.045 | 0.094 | 0.059 | 0.018 | 0.056 | 0.019 | 0.055 | 0.044 | 0.058 |
| SARS-CoV-2 (14,332) | AUC_ROC | 0.599 | **0.700** | 0.686 | 0.688 | 0.649 | 0.643 | 0.638 | 0.646 | 0.640 | 0.683 | 0.651 | 0.663 |
| ESOL (1,127) | RMSE | 0.676 | **0.583** | 0.647 | 2.569 | 1.511 | 0.718 | 0.653 | 0.773 | 0.695 | 0.661 | 0.594 | 0.671 |
| Lipop (4,200) | RMSE | 0.683 | **0.585** | 0.626 | 1.016 | 1.207 | 0.947 | 0.633 | 0.665 | 0.669 | 0.680 | 0.664 | 0.630 |
| FreeSolv (639) | RMSE | 1.063 | **0.715** | 1.014 | 2.275 | 2.205 | 1.504 | 1.046 | 1.316 | 1.327 | 1.304 | 1.139 | 1.159 |
| QM7 (6,830) | MAE | **42.814** | 52.726 | 51.403 | 81.165 | 158.160 | 64.363 | 86.060 | 64.530 | 107.013 | 78.217 | 59.973 | 55.727 |
| QM8 (21,786) | MAE | 0.0364 | 0.0126 | **0.0098** | 0.0205 | 0.0295 | 0.0232 | 0.0104 | 0.0154 | 0.0109 | 0.0187 | **0.0098** | 0.0103 |

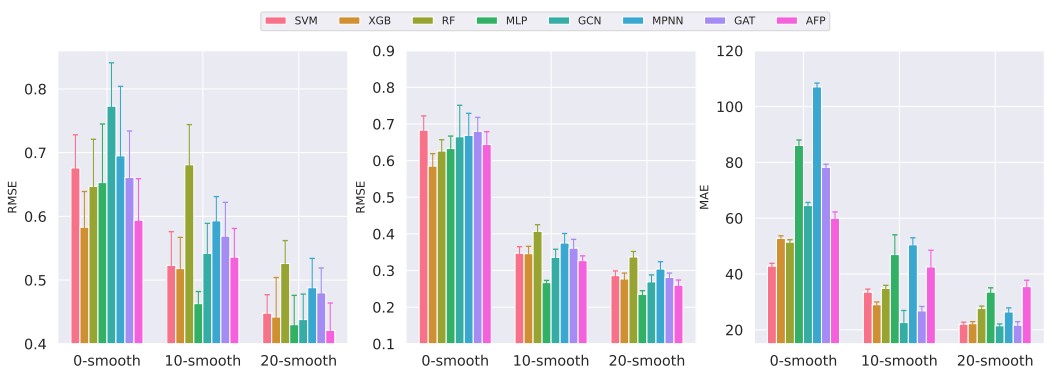

Figure 2: The performance of various models on the smoothed datasets. **Left**: ESOL (Regression); **Middle**: Lipop (Regression); **Right**: QM7 (Regression). Kindly note that we only smooth the regression datasets because the labels of classification datasets are not suitable for smoothing.

MUV and QM 7). However, in some small datasets (e.g., ClinTox and ESOL), some deep models can beat partial non-deep ones. Therefore, we argue that there could be other reasons for the failure of deep models, not solely the dataset size. We will provide further analysis in Sec. 4.

*Observation 3.* **XGB and RF exhibit a particular advantage over other models.**
In the experiments shown in Table 1, we can see that the XGB and RF models consistently rank among the top three on each dataset. Additionally, tree models rank as the top one on 8/15 datasets. Next, we will explore why tree models are well-suited for molecular fingerprints in Sec. 4.

## 4 Empirical Study: Why above phenomena would occur?

In this section, we attempt to understand which characteristics of molecular data lead to the failure of powerful deep models. Also, we aim to understand the inductive biases of XGB and RF that make them well-suited for molecules, and how they differ from the inductive biases of deep models. *The details of the experiments in this section can be found in the appendix.*

*Explanation 1.* **Unlike image data, molecular data patterns are non-smooth. Deep models struggle to learn non-smooth target functions that map molecules to properties.**
We design two experiments to verify the above explanation, i.e., increasing or decreasing the level

Table 2: $\text{RMSE}_c$ are the prediction RMSE on cliff molecules, respectively. $\Delta\mathcal{P}$ is the predicted bioactivity change rate when transitioning from non-cliff to cliff molecules with subtle structural changes. The top-3 performances on each dataset are highlighted with the grey background. The best performance is highlighted with **bold**.

| Target name (Response type) | Metric | SVM | XGB | RF | CNN | RNN | TRSF | MLP | GCN | MPNN | GAT | AFP |
|---|---|---|---|---|---|---|---|---|---|---|---|---|
| CB1 (Agonism $\text{EC}_{50}$) | $\text{RMSE}_c$ | 0.773 | **0.767** | 0.770 | 0.944 | 0.823 | 0.888 | 0.807 | 0.992 | 0.989 | 0.975 | 0.967 |
| | $\Delta\mathcal{P}$ | 15.04% | 21.13% | 20.76% | 2.07% | 10.15% | 9.42% | 13.32% | 4.13% | 3.85% | 1.17% | 4.35% |
| DAT (Inhibition $\text{K}_i$) | $\text{RMSE}_c$ | 0.744 | **0.696** | 0.730 | 0.894 | 0.783 | 0.934 | 0.792 | 1.003 | 0.921 | 1.042 | 0.995 |
| | $\Delta\mathcal{P}$ | 20.64% | 23.03% | 23.95% | 2.73% | 14.18% | 9.34% | 15.02% | 5.83% | 5.15% | 2.27% | 5.08% |
| PPAR$\alpha$ (Agonism $\text{EC}_{50}$) | $\text{RMSE}_c$ | **0.671** | 0.678 | 0.685 | 0.962 | 0.825 | 0.968 | 0.713 | 0.870 | 0.872 | 0.929 | 0.823 |
| | $\Delta\mathcal{P}$ | 21.07% | 22.93% | 23.14% | 11.29% | 13.48% | 18.39% | 15.29% | 1.83% | 5.18% | 4.93% | 11.73% |
| DOR (Inhibition $\text{K}_i$) | $\text{RMSE}_c$ | 0.861 | 0.854 | **0.836** | 1.098 | 1.036 | 1.032 | 0.874 | 1.259 | 1.152 | 1.281 | 1.179 |
| | $\Delta\mathcal{P}$ | 25.26% | 28.41% | 23.95% | 10.02% | 9.83% | 10.25% | 15.18% | 9.77% | 12.52% | 11.36% | 13.11% |

of data smoothing in the molecular datasets. Firstly, we 'increase' the molecular data smoothing level by smoothing the labels based on similarities between molecules. Specifically, let $\mathcal{D}$ denote the molecular dataset and $(x_i, y_i) \in \mathcal{D}$ be $i$-th molecule and its label, we smooth the target function as follows,

$$\widehat{y_i} = \frac{\sum_{x_j \in \mathcal{N}_{x_i}} s(x_i, x_j) y_j}{\sum_{x_j \in \mathcal{N}_{x_i}} s(x_i, x_j)}, \tag{1}$$

where $s(\cdot, \cdot)$ denotes the Tanimoto coefficient of the extended connectivity fingerprints (ECFP) between two molecules that can be considered as their structural similarity. $\mathcal{N}_{x_i}$ is the $k$-nearest neighbor set of $x_i$ (including $x_i$) picked from the whole dataset based on the structural similarities. $\widehat{y_i}$ denotes the label after smoothing. We smooth all the molecules in the dataset in this way and use the smoothed label $\widehat{y_i}$ to train the models. The results are shown in Figure 2, where '0-smooth' denotes the original datasets. '10-smooth' and '20-smooth' mean $k = 10$ and $k = 20$, respectively. As can be observed, the performance of deep models improves dramatically as the level of dataset smoothing increases, and many deep models including MLP, GCN, and AFP can even beat non-deep ones after smoothing. These phenomena indicate that deep models are more suitable for the smoothed datasets.

Secondly, we 'decrease' the level of data smoothing using the concept of activity cliff [40, 60] from chemistry, which means a situation where small changes in the chemical structure of a drug lead to significant changes in its bioactivity. We provide an example activity cliff pairs in Figure 3. Apparently, the target function of activity cliffs that map molecules to the activity values is less smoothing than normal molecular datasets. We then evaluate the models on the activity cliff datasets [66]. The test set contains molecules that are chemically similar to those in the training set but exhibit either a large difference in bioactivity (cliff molecules) or similar bioactivity (non-cliff molecules). As shown in Table 2, the non-deep models consistently outper-

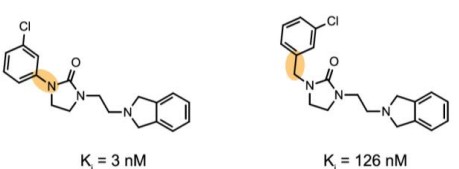

Figure 3: Examplary of Activity Cliffs (ACs) on the target named dopamine D3 receptor (D3R). $\text{K}_i$ means the bioactivity values. The figure is adapted from a previous work [66] with permission.

form deep ones on these activity cliff datasets. Furthermore, the deep models exhibit less significant prediction change rates compared to the non-deep ones. This observation suggests that deep models are indeed less sensitive to subtle structural changes compared to non-deep ones. Our explanation is consistent with the conclusions in deep learning theory [51], i.e., deep models struggle to learn high-frequency components of the target functions. However, traditional models such as XGB and RF can learn piece-wise target functions, and do not exhibit such bias. Our explorations uncover several promising avenues to enhance deep models' performance on molecules: smoothing the target functions or improving deep models' ability to learn the non-smooth target functions.

*Explanation 2.* **Deep models undesirably mix different dimensions of molecular features, whereas tree models make decisions based on each dimension of the features separately.**

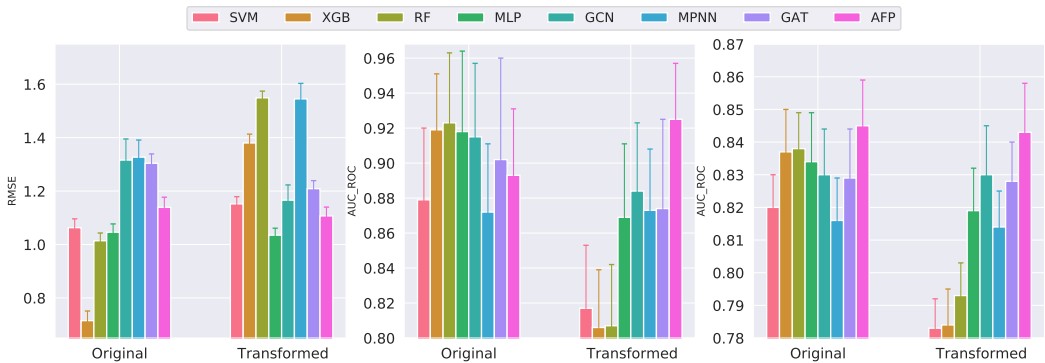

Figure 4: The performance of various models on the orthogonally transformed datasets. **Left**: FreeSolv (Regression); **Middle**: ClinTox (Classification); **Right**: Tox21 (Classification). Kindly note that we did not evaluate CNN, RNN, and TRSF on the transformed datasets because we cannot apply the orthogonal transformations to the input SMILES strings.

Typically, features in molecular data carry meanings individually. As we elaborated in Sec. 2.1, each dimension of molecular fingerprints often indicates whether a certain substructure is present in the molecule; each dimension of nodes/edges features in molecular graph data indicates a specific characteristic of the atoms/bonds (e.g., atom/bond type, atom degree). To verify the above explanation, we mix the different dimensions of molecular features $x_i \in \mathbb{R}^d$ using an orthogonal transformation before feeding them into various models,

$$\widehat{x_i} = \mathcal{Q}x_i, \tag{2}$$

where $\mathcal{Q} \in \mathbb{R}^{d \times d}$ is the orthogonal matrix and $\widehat{x_i}$ is the molecular feature after transformation. Kindly note that the meaning of $x_i$ depends on the input molecular descriptors in the experiments. Specifically, for SVM, XGB, RF, and MLP, $x_i$ denotes the molecular fingerprints; for GNN models, $x_i$ can denote the atom features and bond features in the molecular graphs, i.e., we apply orthogonal transformations to both the atom features and bond features. As can be observed in Figure 4, the performance of tree models deteriorates dramatically and falls behind most deep models after the orthogonal transformation. It is because each dimension of $\widehat{x_i}$ is a linear combination of all the dimensions of $x_i$ according to the matrix-vector product rule. In other words, the molecular features after orthogonal transformation no longer carry meanings individually, accounting for the failure of tree models that make decisions based on each dimension of the features separately. The learning style of tree models is more suitable for molecular data because only a handful of features (e.g., certain substructures) are most indicative of molecular properties [47]. On the other hand, the performance decreases of deep models are less significant, and most deep models can beat tree models after the transformations. We explain this observation as follows. Without the loss of generality, we assume that a linear layer of deep models can map the original molecular feature $x_i$ to the label $y_i$,

$$y_i = W^\top x_i + b, \tag{3}$$

where $W$ and $b$ denote the parameter matrix and the bias term of the linear layer, respectively. And then, we aim to learn a new linear layer mapping the transformed model feature $\widehat{x_i}$ to label $y_i$,

$$y_i = \widehat{W}^\top \widehat{x_i} + b = \widehat{W}^\top \mathcal{Q} x_i + \hat{b}, \tag{4}$$

where $\widehat{W}$ and $\hat{b}$ denote the parameter matrix and the bias term of the new linear layer, respectively. Apparently, to achieve the same results as the original feature, we only have to learn $\widehat{W}$ so that $\widehat{W} = \mathcal{Q}W$ because $\mathcal{Q}^{-1} = \mathcal{Q}^\top$ as an orthogonal matrix, and also $\hat{b} = b$. Therefore, applying the orthogonal transformation to molecular features barely impacts the performance of deep models. The empirical results in Figure 4 confirm this point although some performance changes are observable due to uncontrollable random factors.

## 5 Methodology

Although we have shown and explained the superiority of non-deep models on molecular data, deep models have numerous advantages over the traditional approaches: (i) GNNs can exploit the structural/geometrical inductive biases of 2D/3D molecular graphs, alleviating the manual efforts to capture the topology of the networks; (ii) The pre-trained representations with deep models are beneficial to various downstream tasks, showing promising values in drug discovery [75, 74, 9, 73]. Therefore, we aim to empower the deep models to beat the non-deep ones on MPP tasks including activity cliff cases.

As we explained before, deep models struggle to learn the non-smooth target functions of molecular data, a phenomenon referred to "spectral bias" in literature [51]. To overcome such bias, prior works [44, 87] have experimentally found that a heuristic sinusoidal mapping of the input features allows MLPs to learn the non-smooth target functions. However, these mapping methods would undesirably mix the original features. Please refer to the appendix for detailed discussions due to the limited space. As a remedy, we introduce a new method named Independent Feature Mapping (IFM) that embeds each dimension of molecular features separately before feeding them into models. Denoting a molecular feature as $x \in \mathbb{R}^d$, we formulate IFM as,

$$f_x = [\sin(v) || \cos(v)], \quad v = [2\pi c_1 x, \ldots, 2\pi c_k x],$$ (5)

where $||$ denotes the concatenation of two vectors, $\boldsymbol{c} = [c_1, c_2, \cdots, c_k]$ are the learnable parameters initialized from $\mathcal{N}(0, \sigma)$ and $f_x \in \mathbb{R}^{2k \times d}$. We study the influence of the hyperparameters $k$ and $\sigma$ in the appendix. Since $\cos(a - b) = \cos a \cos b + \sin a \sin b$, we have,

$$f_x \cdot f_{x'} = \sum_{i=1}^{k} \cos(2\pi c_i(x - x')) := g_{\boldsymbol{c}}(x - x'),$$ (6)

where $\cdot$ is the dot product and $x'$ is another molecular feature. Therefore, IFM can map data points to a vector space so that their dot product achieves a certain distance metric, which is an expected characteristic for feature mapping methods [52, 4, 23]. In what follows, we will provide theoretical justifications on the effectiveness of our IFM following a previous study [62]. As revealed in previous works, deep models can be approximated with Neural Tangent Kernel (NTK) [28, 2, 6, 36, 62]. Specifically, let $I$ be a fully-connected deep network with weights $\theta$ initialized from a Gaussian distribution $\mathcal{N}$, the NTK theory shows that as the width of the layers in $I$ becomes infinite and the learning rate for stochastic gradient descent (SGD) approaches zero, the function $I(x; \theta)$ converges during training to the kernel regression solution using the neural tangent kernel (NTK), which is:

$$h_{\mathrm{NTK}}(x, x') = \mathbb{E}_{\theta \sim \mathcal{N}} \left\langle \frac{\partial I(x; \theta)}{\partial \theta}, \frac{\partial I(x'; \theta)}{\partial \theta} \right\rangle$$ (7)

When the inputs are limited to a hypersphere, the NTK for an MLP can be expressed as a dot product kernel (a kernel in the form $h_{\mathrm{NTK}}(x \cdot x')$ for a scalar function $h_{\mathrm{NTK}} : \mathbb{R} \to \mathbb{R}$). In our cases, the input to the deep models would be $f_x$, the composed kernel of IFM and NTK can be formulated as,

$$h_{\mathrm{NTK}}(f_x \cdot f_{x'}) = h_{\mathrm{NTK}}(g_{\boldsymbol{c}}(x - x')) = (h_{\mathrm{NTK}} \circ g_{\boldsymbol{c}})(x - x'),$$ (8)

thus, training deep models on these mapped molecular features corresponds to kernel regression with the stationary composed NTK function $h_{\mathrm{NTK}} \circ g_{\boldsymbol{c}}$. Considering that the parameters $\boldsymbol{c}$ are tunable, IFM creates a composed NTK that is not only stationary but also tunable. It enables us to dramatically control the range of frequencies that can be learned via manipulating the parameters $\boldsymbol{c}$.

## 6 Experiments

### 6.1 Experimental Settings

In our experiments, we equip various deep models with IFM. Specifically, for MLPs with fingerprints as inputs, we employ the proposed feature mapping method to the fingerprints (after feature selection and standardization) directly; For molecular graphs, we map both the features of atoms and bonds before feeding them into the GNNs. The other settings are the same as the benchmarking experiments. If a deep model named 'z' (e.g., MLP) is equipped with IFM, we re-name it as 'IFM-z' (e.g., IFM-MLP) in our results. Also, we evaluate the non-deep models equipped with IFM in the appendix.

Table 3: The performance comparison on multiple molecular datasets. The best performance on each dataset is highlighted with **bold**. The 'P-Best (Model)' denotes the best result in Table 1 and its corresponding model name. The results of 12 tasks on QM 9 can be seen in the appendix.

| Dataset (No.) | Metric | MLP | GCN | MPNN | GAT | AFP | P-Best (Model) | IFM-MLP | IFM-GCN | IFM-MPNN | IFM-GAT | IFM-AFP |
|---|---|---|---|---|---|---|---|---|---|---|---|---|
| BACE (1,513) | AUC_ROC | 0.887 | 0.880 | 0.846 | 0.886 | 0.879 | 0.896 (XGB) | **0.915** | 0.903 | 0.866 | 0.894 | 0.907 |
| HIV (4,0748) | AUC_ROC | 0.791 | 0.834 | 0.814 | 0.812 | 0.819 | 0.834 (GCN) | 0.816 | **0.862** | 0.846 | 0.838 | 0.849 |
| BBBP (2,035) | AUC_ROC | 0.918 | 0.915 | 0.872 | 0.902 | 0.893 | 0.926 (XGB) | 0.937 | **0.945** | 0.908 | 0.933 | 0.940 |
| ClinTox (1,475) | AUC_ROC | 0.890 | 0.889 | 0.868 | 0.891 | 0.907 | **0.963** (TRSF) | 0.941 | 0.938 | 0.929 | 0.953 | 0.959 |
| SIDER (1,366) | AUC_ROC | 0.617 | 0.633 | 0.603 | 0.614 | 0.620 | 0.644 (RF) | 0.646 | 0.649 | 0.638 | 0.647 | **0.652** |
| Tox21 (7,811) | AUC_ROC | 0.834 | 0.830 | 0.816 | 0.829 | 0.845 | 0.845 (AFP) | 0.842 | 0.839 | 0.837 | 0.849 | **0.853** |
| ToxCast (8,539) | AUC_ROC | 0.781 | 0.767 | 0.736 | 0.768 | 0.788 | 0.788 (AFP) | 0.795 | 0.790 | 0.772 | 0.797 | **0.806** |
| MUV (93,087) | AUC_PRC | 0.018 | 0.056 | 0.019 | 0.055 | 0.044 | 0.093 (SVM) | 0.052 | 0.113 | 0.068 | **0.124** | 0.097 |
| SARS-CoV-2 (14,332) | AUC_ROC | 0.638 | 0.646 | 0.640 | 0.683 | 0.651 | 0.700 (XGB) | 0.675 | 0.682 | 0.686 | **0.716** | 0.704 |
| ESOL (1,127) | RMSE | 0.653 | 0.773 | 0.695 | 0.661 | 0.594 | 0.583 (XGB) | 0.587 | 0.728 | 0.673 | 0.566 | **0.561** |
| Lipop (4,200) | RMSE | 0.633 | 0.665 | 0.669 | 0.680 | 0.664 | 0.585 (XGB) | **0.556** | 0.577 | 0.568 | 0.584 | 0.578 |
| FreeSolv (639) | RMSE | 1.046 | 1.316 | 1.327 | 1.304 | 1.139 | **0.715** (XGB) | 0.862 | 0.916 | 0.911 | 0.908 | 0.883 |
| QM7 (6,830) | MAE | 86.060 | 64.530 | 107.013 | 78.217 | 59.973 | 42.814 (SVM) | 66.570 | 38.793 | 84.918 | 59.595 | **33.775** |
| QM8 (21,786) | MAE | 0.0104 | 0.0154 | 0.0109 | 0.0187 | 0.0098 | 0.0098 (AFP) | 0.0091 | 0.0114 | 0.0085 | 0.0139 | **0.0079** |

Table 4: The results on activity cliff datasets. The best result for each dataset is highlighted in '**bold**'.

| Target name (Response type) | Metric | MLP | GCN | MPNN | GAT | AFP | P-Best (Model) | IFM-MLP | IFM-GCN | IFM-MPNN | IFM-GAT | IFM-AFP |
|---|---|---|---|---|---|---|---|---|---|---|---|---|
| CB1 (Agonism $EC_{50}$) | $RMSE_c$ | 0.807 | 0.992 | 0.989 | 0.975 | 0.967 | 0.767 (XGB) | **0.715** | 0.748 | 0.756 | 0.741 | 0.746 |
| DAT (Inhibition $K_i$) | $RMSE_c$ | 0.792 | 1.003 | 0.921 | 1.042 | 0.995 | 0.696 (XGB) | **0.646** | 0.682 | 0.673 | 0.665 | 0.670 |
| PPAR$\alpha$ (Agonism $EC_{50}$) | $RMSE_c$ | 0.713 | 0.870 | 0.872 | 0.929 | 0.823 | 0.671 (SVM) | 0.623 | 0.634 | 0.649 | 0.661 | **0.616** |
| DOR (Inhibition $K_i$) | $RMSE_c$ | 0.874 | 1.259 | 1.152 | 1.281 | 1.179 | 0.836 (RF) | **0.787** | 0.813 | 0.796 | 0.799 | 0.810 |

## 6.2 Results

**Main results.** We show the main results in Table 3, from which we can make the following observations: (1) The proposed feature mapping method can significantly improve the performance of the deep models on molecular datasets; (2) The deep models equipped with the feature mapping method can beat non-deep counterparts in most cases, verifying the effectiveness of our method.

**Results on activity cliffs.** We also employ the proposed feature mapping method on activity cliff datasets where the target functions are less smooth. The results shown in Table 4 indicate that IFM improves the deep models by significant margins. Moreover, nearly all the deep models with IFM method can beat traditional methods, confirming that our method can help neural networks learn non-smooth target functions.

Table 5: Comparisons with SM (Sinusoidal Mapping [44]) and GM (Gaussian Mapping [62]).

| Methods | BACE | HIV | BBBP | ESOL | Lipop | FreeSolv |
|---|---|---|---|---|---|---|
| MLP + SM | 0.890 | 0.798 | 0.925 | 0.640 | 0.613 | 1.033 |
| MLP + GM | 0.896 | 0.806 | 0.923 | 0.625 | 0.591 | 0.992 |
| IFM-MLP | 0.915 | 0.816 | 0.937 | 0.587 | 0.556 | 0.862 |
| GCN + SM | 0.894 | 0.838 | 0.920 | 0.762 | 0.635 | 1.204 |
| GCN + GM | 0.892 | 0.847 | 0.926 | 0.766 | 0.609 | 1.193 |
| IFM-GCN | 0.903 | 0.862 | 0.945 | 0.728 | 0.577 | 0.916 |

Table 6: Pre-training on molecules with encoders equipped with IFM. GIN: Graph Isomorphism Network [78].

| Method (Encoder) | Tox21 | ToxCast | Sider | ClinTox | BBBP | Bace |
|---|---|---|---|---|---|---|
| AttrMasking [27] (GCN) | 0.745 | 0.626 | 0.598 | 0.724 | 0.653 | 0.773 |
| AttrMasking (IFM-GCN) | 0.758 | 0.639 | 0.612 | 0.741 | 0.667 | 0.782 |
| MGSSL [85] (GIN) | 0.752 | 0.633 | 0.616 | 0.771 | 0.688 | 0.788 |
| MGSSL (IFM-GIN) | 0.764 | 0.639 | 0.623 | 0.783 | 0.704 | 0.796 |
| Mole-BERT [26] (GIN) | 0.768 | 0.643 | 0.628 | 0.789 | 0.719 | 0.808 |
| Mole-BERT (IFM-GIN) | 0.775 | 0.649 | 0.627 | 0.796 | 0.724 | 0.813 |

**Comparisons with other feature mapping methods (ablation study).** We compare the proposed IFM with the previous feature mapping methods. The results shown in Table 5 indicate that the proposed feature mapping method is superior to the previous method, which verifies that mixing different dimensions of molecular features as GM [62] would degrade the performance.

**Pre-training on molecules with IFM.** Compared with non-deep models, deep ones can be combined with the prevalent 'Pretraining and Finetuning' paradigm to exploit large-scale unlabeled molecules [75]. This motivates us to develop more powerful neural encoders for this paradigm. Specifically, we pre-train the deep models equipped with IFM and report the fine-tuning results in Table 6. Our experimental settings are the same as the pioneering work [27] where the datasets are split with scaffold splitting, differing from the random splitting in Table 3. As can be observed, our method can boost various pre-training strategies to advance their performance in downstream tasks.

# 7 Discussion and Conclusion

In this paper, we perform a comprehensive benchmark of representative models on molecular property prediction. Our results reveal that traditional machine learning models, especially tree models, can easily outperform well-designed deep models in most cases. These phenomena can be attributed to the unique patterns of molecular data and different inductive biases of various models. Specifically, the target function mapping molecules to properties are non-smooth, and some small changes can incur significant property variance. Deep models struggle to learn such patterns. Additionally, molecular features carry meanings individually and deep models would undesirably mix different dimensions of molecular features. These findings stimulate us to develop a simple-yet-effective feature mapping method for molecule data that can help deep models learn non-smooth target functions with theoretical guarantees. Extensive experiments verify the effectiveness of the proposed method. Our study leaves an open question for future research: Can our findings and methods be generalized to other AIDD tasks including drug-target interactions (DTIs) prediction, drug-drug interactions (DDIs) prediction?

# 8 Acknowledgements

We thank the anonymous the area chairs and reviewers for their constructive and helpful reviews. This work was supported by the National Key R&D Program of China (Project 2022ZD0115100), the National Natural Science Foundation of China (Project U21A20427), the Research Center for Industries of the Future (Project WU2022C043), and the Competitive Research Fund (Project WU2022A009) from the Westlake Center for Synthetic Biology and Integrated Bioengineering.

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

# A  Datasets

Table 7: Summary for the molecule datasets in the benchmark.

| Dataset | Task | # Tasks | # Molecules |
|---------|------|---------|-------------|
| BBBP | Classification | 1 | 2,039 |
| Tox21 | Classification | 12 | 7,831 |
| ToxCast | Classification | 617 | 8,576 |
| Sider | Classification | 27 | 1,427 |
| ClinTox | Classification | 2 | 1,478 |
| MUV | Classification | 17 | 93,087 |
| HIV | Classification | 1 | 41,127 |
| Bace | Classification | 1 | 1,513 |
| SARS-CoV-2 | Classification | 13 | 14,332 |
| Lipop | Regression | 1 | 4,200 |
| FreeSolv | Regression | 1 | 642 |
| ESOL | Regression | 1 | 1,1128 |
| QM7 | Regression | 1 | 7,160 |
| QM8 | Regression | 16 | 21,786 |
| QM9 | Regression | 12 | 133,885 |

In this section, we provide detailed information on the molecular datasets used for downstream tasks.
**Molecular Property: Pharmacology**  The Blood-Brain Barrier Penetration (BBBP) dataset measures whether a molecule will penetrate the central nervous system. All three datasets, Tox21 [1], ToxCast [72], and ClinTox [19] are related to the toxicity of molecular compounds. The Side Effect Resource (SIDER) dataset stores the adverse drug reactions on a marketed drug database.
**Molecular Property: Physical Chemistry**  Dataset proposed in [12] measures aqueous solubility of the molecular compounds. Lipophilicity (Lipo) dataset is a subset of ChEMBL [18] measuring the molecule octanol/water distribution coefficient. CEP dataset is a subset of the Havard Clean Energy Project (CEP), which estimates the organic photovoltaic efficiency.
**Molecular Property: Biophysics**  Maximum Unbiased Validation (MUV) is another sub-database from PCBA, and is obtained by applying a refined nearest neighbor analysis. HIV is from the Drug Therapeutics Program (DTP) AIDS Antiviral Screen, and it aims at predicting inhibit HIV replication. BACE measures the binding results for a set of inhibitors of $\beta$-secretase 1 (BACE-1) and is gathered in MoleculeNet [72].
**Molecular Property: Quantum Mechanics**  QM7, QM8 and QM9 contain stable organic molecules with up to 7, 8 or 9 heavy atoms. 3D atomic coordinates as well as electrical properties of molecules were calculated with ab initio Density Functional Theory (DFT).
**Molecular Property: COVID-19**  SARS-CoV-2 [7] is a collection of datasets[3] generated by screening a panel of SARS-CoV-2-related assays against approved drugs. 13 assays of 14,332 drugs were used in our experiments.
Statistics of the datasets are in Table 7.

---

[3]The datasets are available at https://opendata.ncats.nih.gov/covid19/ (CC BY 4.0 license) and are continuously extended. The data used in our experiments were downloaded on February 16th, 2021.

# B Results on QM9 dataset with 12 tasks

We show the results of each task of QM9 dataset in Table 1. As can be observed, the proposed IFM module can consistently improve the performance of base models and can even outperform the SPN with 3D geometry information as inputs in most tasks.

Table 8: Comparison of MAE on QM9. The best result is highlighted with '**bold**'.

| Target Unit | $\epsilon_{HOMO}$ eV | $\epsilon_{LUMO}$ eV | $\Delta\epsilon$ eV | $\mu$ D | $\alpha$ bohr$^3$ | $R^2$ $a_0^2$ | ZPVE meV | $U_0$ meV | $U$ meV | $H$ meV | $G$ meV | $c_v$ cal/molK |
|---|---|---|---|---|---|---|---|---|---|---|---|---|
| SVM | 0.016 | 0.038 | 0.040 | 1.19 | 6.35 | - | 0.026 | - | - | - | - | 3.22 |
| XGB | 0.016 | 0.039 | 0.040 | 1.19 | 6.36 | - | 0.026 | - | - | - | - | 3.22 |
| RF | 0.016 | 0.038 | 0.039 | 1.19 | 6.34 | - | 0.020 | - | - | - | - | 3.22 |
| CNN | 0.018 | 0.019 | 0.032 | 1.11 | 1.55 | 99.27 | 0.073 | 3.52 | 4.09 | 3.81 | 4.03 | 1.08 |
| RNN | 0.014 | 0.028 | 0.028 | 0.91 | 3.53 | 91.03 | 0.015 | 14.83 | 14.93 | 15.66 | 14.84 | 1.47 |
| TRSF | 0.011 | 0.015 | 0.017 | 0.83 | 1.46 | 96.67 | 0.057 | 1.97 | 2.95 | 1.90 | 1.73 | 0.92 |
| GCN | 0.030 | 0.015 | 0.019 | 0.69 | 0.77 | 27.49 | 0.014 | 0.55 | 0.56 | 0.55 | 0.55 | 0.34 |
| MPNN | 0.100 | 0.440 | 0.410 | 0.95 | 1.47 | 38.96 | 0.410 | 2.15 | 2.19 | 2.16 | 2.18 | 0.93 |
| GAT | 0.060 | 0.012 | 0.016 | 0.69 | 0.85 | 24.85 | 0.057 | 0.79 | 0.81 | 0.80 | 0.80 | 0.43 |
| AFP | 0.089 | 0.029 | 0.073 | 0.90 | 0.74 | 27.35 | 0.110 | 1.20 | 1.18 | 1.19 | 1.19 | 0.51 |
| SPN | 0.024 | 0.019 | 0.032 | **0.31** | **0.47** | 29.26 | **0.011** | 0.63 | 0.73 | 0.64 | 0.82 | **0.20** |
| IFM-MLP | 0.013 | 0.022 | 0.032 | 0.87 | 1.04 | 125.47 | 0.015 | 16.19 | 16.23 | 16.20 | 16.22 | 0.97 |
| IFM-GCN | 0.019 | 0.011 | 0.014 | 0.63 | 0.53 | 15.18 | 0.012 | **0.46** | **0.43** | **0.47** | **0.44** | 0.25 |
| IFM-MPNN | 0.076 | 0.371 | 0.358 | 0.77 | 0.94 | 19.82 | 0.252 | 1.88 | 1.86 | 1.86 | 1.87 | 0.75 |
| IFM-GAT | 0.032 | **0.009** | **0.010** | 0.53 | 0.69 | 20.15 | 0.038 | 0.62 | 0.63 | 0.62 | 0.61 | 0.29 |
| IFM-AFP | **0.009** | 0.017 | 0.575 | 0.74 | 0.48 | **14.69** | 0.096 | 0.86 | 0.88 | 0.87 | 0.86 | 0.33 |

# C The influence of the hyperparameters $k$ and $\sigma$ in IFM

Table 9: The influence of the hyper-parameter $k$ (with $\sigma = 6$).

| Models / $k$ | 4 | 8 | 16 | 32 | 64 |
|---|---|---|---|---|---|
| Toxcast (IFM-MLP) | 0.782 | 0.795 | 0.797 | 0.789 | 0.792 |
| Toxcast (IFM-GCN) | 0.771 | 0.780 | 0.783 | 0.785 | 0.774 |
| Sider (IFM-MLP) | 0.625 | 0.631 | 0.626 | 0.635 | 0.636 |
| Sider (IFM-GCN) | 0.635 | 0.638 | 0.639 | 0.633 | 0.632 |

Table 10: The influence of the hyper-parameter $\sigma$ (with $k = 8$).

| Models / $\sigma$ | 1 | 3 | 6 | 9 | 12 | 15 |
|---|---|---|---|---|---|---|
| Toxcast (IFM-MLP) | 0.767 | 0.782 | 0.795 | 0.791 | 0.785 | 0.772 |
| Toxcast (IFM-GCN) | 0.763 | 0.775 | 0.780 | 0.790 | 0.788 | 0.776 |
| Sider (IFM-MLP) | 0.611 | 0.626 | 0.631 | 0.646 | 0.639 | 0.613 |
| Sider (IFM-GCN) | 0.620 | 0.633 | 0.638 | 0.649 | 0.643 | 0.627 |

$k$ and $\sigma$ are hyper-parameters in IFM and they are tuned for the optimal performance using the validation set. We show their tunning space and corresponding results in Table 9 and Table 10, respectively. As can be observed, the hyper-parameter $k$ barely impacts the performance of IFM. Larger $k$ would undesirably incur more computational overhead. In contrast, smaller or larger $\sigma$ would degrade the performance of IFM dramatically. In other words, the hyper-parameter $\sigma$ is more important and we are encouraged to set larger tuning space for $\sigma$ to pick the optimal value for each dataset using the validation set.

## D Performance of Non-deep models (SVM, RF, and XGB) equipped with IFM

In Table 11, we also show the results of non-deep models (SVM, RF, and XGB) equipped with IFM. As can be observed, IFM barely impacts the performance of non-deep models.

Table 11: Performance of Non-deep models (SVM, RF, and XGB) equipped with IFM

| Models | Tox21 | ToxCast | Sider | ClinTox | BBBP | Bace |
|---|---|---|---|---|---|---|
| SVM | 0.820 | 0.725 | 0.626 | 0.879 | 0.913 | 0.886 |
| IFM-SVM | 0.822 | 0.724 | 0.629 | 0.883 | 0.907 | 0.883 |
| RF | 0.838 | 0.778 | 0.644 | 0.933 | 0.923 | 0.890 |
| IFM-RF | 0.843 | 0.785 | 0.643 | 0.935 | 0.929 | 0.883 |
| XGB | 0.837 | 0.785 | 0.638 | 0.919 | 0.926 | 0.896 |
| IFM-XGB | 0.833 | 0.780 | 0.636 | 0.911 | 0.933 | 0.894 |

## E The performance of CNN, RNN, and TRSF on smoothed molecular datasets

Due to the limited space in the main text, we report the results of CNN, RNN, and TRSF on smoothed molecular datasets in Table 12 here.

Table 12: The performance of CNN, RNN, and TRSF on smoothed molecular datasets.

| | ESOL | | | Lipop | | | FreeSolv | | |
|---|---|---|---|---|---|---|---|---|---|
| Smoothing level | 0-smooth | 10-smooth | 20-smooth | 0-smooth | 10-smooth | 20-smooth | 0-smooth | 10-smooth | 20-smooth |
| CNN | 2.569 | 1.937 | 1.334 | 1.016 | 0.524 | 0.409 | 2.275 | 1.592 | 0.986 |
| RNN | 1.511 | 1.074 | 0.862 | 1.207 | 0.770 | 0.687 | 2.205 | 1.570 | 1.028 |
| TRSF | 0.718 | 0.667 | 0.564 | 0.947 | 0.549 | 0.435 | 1.504 | 1.175 | 0.954 |

## F Standard deviations of the benchmarking results and main results

For the limited space in the main text, we report the standard deviations of the benchmarking results and main results here in Table 13 and Table 14, respectively.

## G Discussions on the previous feature mapping methods

**Gaussian Feature Mapping [62] (GM):** Given the molecular feature as $x \in \mathbb{R}^d$, $f_x = [\cos(2\pi \mathbf{B}x), \sin(2\pi \mathbf{B}x)]^{\mathrm{T}}$, where each entry in $\mathbf{B} \in \mathbb{R}^{m \times d}$ is sampled from $\mathcal{N}(0, \sigma^2)$. In our experiments, we use an isotropic Gaussian distribution.

**Sinusoidal Feature Mapping (SM):** The general form of Sinusoidal Feature Mapping in previous works [44, 87] can be formulated as $f_x = [\ldots, \cos(2\pi \sigma^{j/m} x), \sin(2\pi \sigma^{j/m} x), \ldots]^{\mathrm{T}}$ for $j = 0, \cdots, m - 1$.

**The superiority of IFM over SM and GM:** The matrix-vector product in GM would undesirably mix different dimensions of molecular features, our IFM avoids this issue. Moreover, the parameters $\mathbf{c}$ in IFM is learnable while the parameters $\mathbf{B}$ are sampled from a fixed Gaussian distribution. Additionally, the feature mapping in SM is deterministic and only contains on-axis frequencies, making it naturally biased towards data that has more frequency content along the axes. In contrast, our IFM allows all directions to share the same frequency content.

Table 13: The standard deviations of main results.

| Dataset (No.) | Metric | IFM-MLP | IFM-GCN | IFM-MPNN | IFM-GAT | IFM-AFP |
|---|---|---|---|---|---|---|
| BACE (1,513) | AUC_ROC | 0.024 | 0.028 | 0.022 | 0.023 | 0.025 |
| HIV (40,748) | AUC_ROC | 0.022 | 0.027 | 0.029 | 0.033 | 0.032 |
| BBBP (2,035) | AUC_ROC | 0.026 | 0.029 | 0.022 | 0.032 | 0.035 |
| ClinTox (1,475) | AUC_ROC | 0.049 | 0.042 | 0.040 | 0.047 | 0.046 |
| SIDER (1,366) | AUC_ROC | 0.025 | 0.027 | 0.028 | 0.025 | 0.029 |
| Tox21 (7,811) | AUC_ROC | 0.017 | 0.014 | 0.015 | 0.012 | 0.011 |
| ToxCast (8,539) | AUC_ROC | 0.017 | 0.016 | 0.020 | 0.019 | 0.017 |
| MUV (93,087) | AUC_PRC | 0.028 | 0.032 | 0.018 | 0.019 | 0.023 |
| SARS-CoV-2 (14,332) | AUC_ROC | 0.027 | 0.025 | 0.020 | 0.018 | 0.023 |
| ESOL (1,127) | RMSE | 0.082 | 0.063 | 0.088 | 0.065 | 0.063 |
| Lipop (4,200) | RMSE | 0.036 | 0.073 | 0.051 | 0.038 | 0.046 |
| FreeSolv (639) | RMSE | 0.207 | 0.247 | 0.258 | 0.261 | 0.259 |
| QM7 (6,830) | MAE | 0.025 | 0.028 | 0.022 | 0.013 | 0.024 |
| QM8 (21,786) | MAE | 0.0002 | 0.0003 | 0.0012 | 0.0003 | 0.0003 |

Table 14: The standard deviations of benchmarking results.

| Dataset (No.) | Metric | SVM | XGB | RF | CNN | RNN | TRSF | MLP | GCN | MPNN | GAT | AFP | SPN |
|---|---|---|---|---|---|---|---|---|---|---|---|---|---|
| BACE (1,513) | AUC_ROC | 0.018 | 0.020 | 0.023 | 0.030 | 0.009 | 0.007 | 0.025 | 0.021 | 0.020 | 0.026 | 0.022 | 0.021 |
| HIV (40,748) | AUC_ROC | 0.019 | 0.020 | 0.017 | 0.009 | 0.029 | 0.011 | 0.019 | 0.024 | 0.029 | 0.033 | 0.028 | 0.030 |
| BBBP (2,035) | AUC_ROC | 0.029 | 0.025 | 0.027 | 0.026 | 0.019 | 0.014 | 0.028 | 0.026 | 0.032 | 0.034 | 0.030 | 0.033 |
| ClinTox (1,475) | AUC_ROC | 0.041 | 0.042 | 0.039 | 0.008 | 0.098 | 0.007 | 0.048 | 0.045 | 0.043 | 0.053 | 0.042 | 0.046 |
| SIDER (1,366) | AUC_ROC | 0.020 | 0.019 | 0.017 | 0.012 | 0.015 | 0.009 | 0.026 | 0.027 | 0.024 | 0.029 | 0.025 | 0.024 |
| Tox21 (7,811) | AUC_ROC | 0.008 | 0.011 | 0.013 | 0.004 | 0.007 | 0.004 | 0.015 | 0.018 | 0.014 | 0.013 | 0.012 | 0.014 |
| ToxCast (8,539) | AUC_ROC | 0.007 | 0.005 | 0.007 | 0.006 | 0.010 | 0.005 | 0.018 | 0.017 | 0.020 | 0.021 | 0.018 | 0.019 |
| MUV (93,087) | AUC_PRC | 0.039 | 0.026 | 0.030 | 0.014 | 0.017 | 0.060 | 0.027 | 0.035 | 0.030 | 0.015 | 0.022 | 0.026 |
| SARS-CoV-2 (14,332) | AUC_ROC | 0.110 | 0.010 | 0.013 | 0.007 | 0.023 | 0.015 | 0.028 | 0.028 | 0.019 | 0.007 | 0.025 | 0.017 |
| ESOL (1,127) | RMSE | 0.049 | 0.051 | 0.073 | 0.107 | 0.043 | 0.048 | 0.089 | 0.057 | 0.092 | 0.068 | 0.067 | 0.073 |
| Lipop (4,200) | RMSE | 0.037 | 0.032 | 0.029 | 0.016 | 0.009 | 0.020 | 0.033 | 0.080 | 0.053 | 0.041 | 0.039 | 0.035 |
| FreeSolv (639) | RMSE | 0.165 | 0.180 | 0.236 | 0.117 | 0.114 | 0.098 | 0.212 | 0.255 | 0.263 | 0.269 | 0.271 | 0.259 |
| QM7 (6,830) | MAE | 0.110 | 0.010 | 0.013 | 0.023 | 0.033 | 0.035 | 0.028 | 0.028 | 0.019 | 0.007 | 0.025 | 0.027 |
| QM8 (21,786) | MAE | 0.0001 | 0.0002 | 0.0002 | 0.0004 | 0.0001 | 0.0004 | 0.0002 | 0.0005 | 0.0014 | 0.0004 | 0.0002 | 0.0003 |

## H   Hardware

For all our benchmarks and experiments, we use the GPUs including NVIDIA V100, and NVIDIA A100 GPUs based on availability.

## I   Hyper-parameters tuning space

For **SVM**, we adopt the widely-used radial basis function (RBF) as the kernel and we optimize the hyper-parameters within the following ranges: $C$ (0.1 to 100) and gamma values (0 to 0.2).

For **XGB**, we penalize the complexity of the traditional gradient boosting model and perform shrinkage and column subsampling to prevent over-fitting. Also, we employ sparsity-aware split finding technique for efficient training on sparse data, etc. In the experiments, we tune the hyper-parameters within the following ranges: learning_rate (0.01 to 0.2), gamma (0 to 0.2), min_child_weight (1 to 6), subsample (0.7 to 1.0), colsample_bytree (0.7 to 1.0), max_depth (3 to 10) and n_estimators (50, 100, 200, 300, 400, 500, 1000).

For **RF**, in the implementation of RF algorithm, sample perturbation via bootstrap sampling of the training data and feature perturbation via random feature subset selection are introduced to improve the diversity of base learners (decision trees), which corrects for the overfitting habit of decision trees and subsequently enhances the generalization ability of RF. In the experiments, we optimize the

hyper-parameters within the following ranges: n_estimators (50, 100, 200, 300, 400, 500, 1000), max_depth (3 to 12), min_samples_leaf (1,3,5,10,20,50), min_impurity_decrease (0 to 0.01) and max_features ('sqrt', 'log2', 0.7, 0.8, 0.9).

For **1D CNN**, we optimize the hyper-parameters within the following ranges: learning rate ($5 \times 10^{-4}$, $5 \times 10^{-5}$, $5 \times 10^{-6}$), the convolution kernel size (4, 8, 10), and the number of the hidden features in the fully connected layer (128, 256, 512, 1024).

For **RNN**, we optimize the hyper-parameters within the following ranges: the hidden dimensions of each layer ([64, 128, 256, 512]), $L_2$ regularization (0 to 0.01), dropout rate (0 to 0.5). The other hyperparameters are the same as the previous work [25].

For **TRSF**, we optimize the hyper-parameters within the following ranges: the hidden dimensions of each layer ([64, 128, 256, 512]), $L_2$ regularization (0 to 0.01), dropout rate (0 to 0.5). The other hyperparameters are the same as the previous work [25].

For **MLP**, we optimize the hyper-parameters within the following ranges: $L_2$ regularization (0 to 0.01), dropout rate (0.0 to 0.5), and neurons for each hidden layer (64, 128, 256, 512). The other important hyper-parameters were fixed: ReLU function that has been recommended by many previous studies was used as the activation function, and the optimizer was set to an adaptive learning rate algorithm: Adadelta [83].

For **MPNN**, we optimize the hyper-parameters within the following ranges: $L_2$ regularization (0, $10 \times 10^{-8}$, $10 \times 10^{-6}$, $10 \times 10^{-4}$), learning rate ($10 \times 10^{-2.5}$, $10 \times 10^{-3.5}$, $10 \times 10^{-1.5}$), dimension of node feature in hidden layers (64, 32, 16), dimension of edge feature in hidden layers (64, 32, 16), and the number of set2set layers (2,3,4). The number of message passing steps and set2set steps were fixed to 6.

For **GCN**, we optimize the hyper-parameters within the following ranges: $L_2$ regularization (0, $10 \times 10^{-8}$, $10 \times 10^{-6}$, $10 \times 10^{-4}$), learning rate ($10 \times 10^{-2.5}$, $10 \times 10^{-3.5}$, $10 \times 10^{-1.5}$), dimension of the FNN classifier (64, 128, 256), and the dimension of GCN hidden layers ([128, 128], [256, 256], [128, 64], [256, 128]).

For **GAT**, we optimize the hyper-parameters within the following ranges: $L_2$ regularization (0, $10 \times 10^{-8}$, $10 \times 10^{-6}$, $10 \times 10^{-4}$), learning rate ($10 \times 10^{-2.5}$, $10 \times 10^{-3.5}$, $10 \times 10^{-1.5}$), dimension of GAT hidden layers ([128, 128], [256, 256], [128, 64], [256, 128]), dimension of FNN classifier (64, 128, 256), and the number of attention heads ([2,2], [3,3], [4,4], [3,4], [2,3]).

For **AFP**, we optimize the hyper-parameters within the following ranges: $L_2$ regularization (0, $10 \times 10^{-8}$, $10 \times 10^{-6}$, $10 \times 10^{-4}$), learning rate ($10 \times 10^{-2.5}$, $10 \times 10^{-3.5}$, $10 \times 10^{-1.5}$), the number of attentive layers for atom embedding (2,3,4,5,6), the number of attentive layers for molecule embedding (1,2,3,4,5), dropout rate (0, 0.1, 0.3, 0.5), and fingerprint dimension (50, 100, 200, 300).

For **SPN**, we optimize the hyper-parameters within the following ranges: the distance LB2 intermediate size (4, 8, 16), the angle LB2 intermediate size (4, 8, 16), the torsion LB2 intermediate size (4, 8, 16), the number of interaction blocks (3, 4, 5), the number of spherical harmonics (3, 5, 7), the cutoff distance (4, 5, 6), the batch size (32, 64), the initial learning rate ($1 \times 10^{-4}$, $5 \times 10^{-4}$, $1 \times 10^{-3}$), the learning rate step size (50, 100, 150), the learning rate decay ratio (0.4, 0.5, 0.6). The other settings are the same as the original paper [38].

