# OpenReview forum: "Understanding the Limitations of Deep Models for Molecular property prediction: Insights and Solutions"
_NeurIPS.cc/2023/Conference — NeurIPS 2023 poster_

### Official Review · Reviewer_SHsm · 2023-07-04

**Soundness:** 3 good
**Presentation:** 3 good
**Contribution:** 2 fair
**Rating:** 5
**Confidence:** 5

**Summary:**

The authors systematically study the performance of deep vs. non-deep machine-learned models in the setting of molecular property prediction, assessing the impact of input representations and models on 15 representative datasets taken from MoleculeNet, a standard benchmark in molecular property prediction.

The authors undertake a study into why non-deep models perform well on these molecular benchmark datasets, including experimentation with smoothing molecular data, and using activity cliff pairs as an example of non-smooth data. There is discussion of two possible explanations as to why non-deep models perform well in molecular property prediction, focusing on smoothness of data and mixing of input features.

**Strengths:**

The strength of this paper lies in its exploration of possible explanations for poor performance of deep-learning approaches in molecular property prediction, as this has not been systematically addressed previously. The exploration of the smoothness of the target function is a potentially valuable addition to the understanding of the field. Discussion is clear and concise, and experiments useful. The explanations for poor performance of non-deep models are plausible and well-supported by the experiments. It is helpful that the authors focused on the MoleculeNet benchmark as this is a good standard in the community.

**Weaknesses:**

In the related work section, there is a good description of the different approaches to representation and modelling of molecular properties. However, this might be better termed "background" or introduction necessary to understand the methods benchmarked rather than related work.  There is no discussion of works actually related to the topic of the paper -- that is, previous work that attempts to benchmark a variety of methods -- such as the MoleculeNet paper itself (Wu et al., 2018) and works building on that which include a variety of non-deep methods, and the range of few-shot learning works in molecular property prediction that also explore the MoleculeNet benchmark datasets (Nguyen et al. 2020 ICML GRL+, Pappu et al. 2020 CoRR), as well as similar works exploring performance of non-deep methods on small property prediction datasets (eg. Stanley et al. 2021).

Therefore, the authors must include a thorough discussion of these works, and the current results, including those reported on the MoleculeNet website for a range of non-deep techniques such as xgb.

At the beginning of section 4, the authors assert that "unlike image data, molecular data patterns are non-smooth" -- this is a somewhat strong assertion to make without evidence, particular as the authors then go on to smooth data with regression labels, which inherently images are not. This statement needs some clarification; overall, it would be useful to more tightly define what is meant by "smooth" data in this section.

In addition, the authors mention pretraining on molecules with IFM and note that it boosts the performance of deep models to use a pretraining and finetuning and pretraining paradigm. However it is not clear how this fits with the explanations given for the poor performance of such models in non-smooth datasets. Do the authors claim that this pretraining is somehow reducing the requirement for smooth data? Clarification and experimentation around this point would be useful.

Overall while the paper attempts to address the possible explanations for poor performance of deep models in molecular property prediction, a major weakness is that it is hard to conclusively state that these reasons are in fact absolute truth, especially given that there are a number of other issues in that domain that are equally pressing (eg. the small size of datasets from a deep-learning perspective, which the authors do not address). The ideas are helpful and explored well but inconclusive.

**Questions:**

As stated above, the authors mention pretraining on molecules with IFM and note that it boosts the performance of deep models to use a pretraining and finetuning and pretraining paradigm. However it is not clear how this fits with the explanations given for the poor performance of such models in non-smooth datasets. Do the authors claim that this pretraining is somehow reducing the requirement for smooth data? Clarification and experimentation around this point would be useful.

**Limitations:**

Limitations of the work are not discussed.

---

> ### Author Response · Authors · 2023-08-16
> **Response to Reviewer SHsm (1/2)**
>
> > *There is no discussion of works actually related to the topic of the paper -- that is, previous work that attempts to benchmark a variety of methods.*
>
> Thanks for your valuable feedback! We have discussed previous works that benchmark various models in Line 42 - 46. Specifically, these studies neither consider the emerging powerful deep  models (e.g., Transformer, SphereNet[1]) nor explore various molecular descriptors (e.g., 3D molecular graph). More importantly, they did not investigate the reasons why deep models often fail on molecules. Also, we would add the following discussions on the two works you recommended: Nguyen et al. [2] utilize meta-learning to improve the deep models' performance in low-resource drug discovery. However, they did not evaluate their method on the widely used MoleculeNet benchmark; Wu et al.[3] evaluate various models on the MoleculeNet benchmark datasets. However, they did not evaluate more advanced models. Additionally, could you kindly provide the specific titles of the other two works (Pappu et al. 2020 CoRR & Stanley et al. 2021) because we cannot find them online only with your prompts? We would like to discuss these works in the future version. Also, following your valuable advice, we would discuss and compare these works in the related works section. Thanks for your helpful reviews again!
>
> > *At the beginning of section 4, the authors assert that "unlike image data, molecular data patterns are non-smooth" This statement needs some clarification; overall, it would be useful to more tightly define what is meant by "smooth" data in this section.*
>
> Thanks for your helpful advice! "Molecular data patterns are non-smooth" means that the properties of molecule data would change a lot with small structural variations (such as replacing an atom or bond). In other words, the target function mapping molecules to properties is un-smooth, i.e., small changes in the inputs would alter the model outputs dramatically. In contrast, in image data, changing a few pixels usually does not completely alter the image category. We apologize for providing a conclusion at the beginning of Section 4 while verifying it subsequently. The writing style may confuse the readers. Based on your valuable advice, we will make sure to define "smooth" clearly in Section 4 in future versions of the manuscript.
>
> > *The authors mention pretraining on molecules with IFM and note that it boosts the performance of deep models to use a pretraining and finetuning paradigm. However, it is not clear how this fits with the explanations given for the poor performance of such models in non-smooth datasets. Do the authors claim that this pretraining is somehow reducing the requirement for smooth data? Clarification and experimentation around this point would be useful.*
>
> Thanks for your helpful reviews! To verify that the proposed IFM can work well in various settings, we evaluate IFM in the pretraining and finetuning paradigm considering that this paradigm has gained great popularity in molecular property prediction recently. The purpose of these experiments is to validate that IFM can help deep models learn the un-smooth target functions mapping molecules to properties in various settings because the downstream task of the paradigm is also molecule property prediction. We believe that these results will make our results more comprehensive.
>
>
> [1] Spherical Message Passing for 3D Graph Networks (ICLR 2022)
> [2] Meta-Learning GNN Initializations for Low-Resource Molecular Property Prediction (ICML 2020 GRL+)
> [3] MoleculeNet: a benchmark for molecular machine learning (Chemical Science 2018)

---

> > ### Author Response · Authors · 2023-08-16
> > **Response to Reviewer SHsm (2/2)**
> >
> > > *A  weakness is that it is hard to conclusively state that these reasons are in fact absolute truth, especially given that the small size of datasets from a deep-learning perspective in that domain that are equally pressing. The authors do not address. *
> >
> > **Table Re1**: The results (mean $\pm$ std) of representative models on OGB-molhiv (ROC-AUC score) and OGB-molpcba (the average precision score). The top-3 performance for each dataset is highlighted with **bold**.
> > | Models| SVM |XGB|RF| CNN| RNN| TRSF| MLP| GCN| MPNN| GAT| D-MPNN| AFP | SPN|
> > |----|----|----|----|----| ----  |---- |----|----|----| ----  |---- |---- |----|
> > | OGB-molhiv (41,127) | 0.8253 $\pm$ 0.0036| **0.8297** $\pm$ 0.0051 | **0.8329** $\pm$ 0.0023 | 0.7915 $\pm$ 0.0083 | 0.7836 $\pm$ 0.0040| 0.8135 $\pm$ 0.0062| 0.8265 $\pm$ 0.0052| 0.8193 $\pm$ 0.0078| 0.8026 $\pm$ 0.0053| 0.8117 $\pm$ 0.0040|**0.8276** $\pm$ 0.0092| 0.8247 $\pm$ 0.0074|0.8018 $\pm$ 0.0067|
> > | OGB-molpcba (437,929)| 0.2963 $\pm$ 0.0035| 0.3054 $\pm$ 0.0027 | **0.3072** $\pm$ 0.0019| 0.2547 $\pm$ 0.0014 | 0.2832 $\pm$ 0.0026 | 0.2759 $\pm$ 0.0033| **0.3058** $\pm$ 0.0026| 0.2811 $\pm$ 0.0023| 0.2785 $\pm$ 0.0019| 0.2816 $\pm$ 0.0037|0.3016 $\pm$ 0.0037| **0.3125** $\pm$ 0.0024|0.2955 $\pm$ 0.0038|
> >
> > Thanks for your helpful reviews! We evaluated 13 models on two larger molecule datasets OGB-molhiv (with 41,127 molecules) and OGB-molpcba (with 437,929 molecules) in Table Re1. As can be observed, non-deep models can also beat most deep models on the large molecule dataset (OGB-molpcba). Furthermore, we would like to highlight that the public leaderboard (available online) in OGB demonstrates that a combination of one molecule fingerprint and random forest can outperform many deep models. While it is true that some deep models achieve better performance on the leaderboard, they often rely on additional data, large-scale pre-training, or ensemble learning techniques.
> >
> > Additionally, in our paper (Line 178 - 180), we observe that all the non-deep models can outperform any deep ones on some larger-scale datasets (e.g., MUV and QM 7). However, in some small datasets (e.g., ClinTox and ESOL), some deep models can beat partial non-deep ones.
> >
> > Therefore, we argue that there is indeed another important factor contributing to the limitations of deep models, beyond the small size of molecule datasets. Moreover, it is worth noting that small-scale molecule datasets are more prevalent in practice, as labeling molecules typically relies on labor-intensive wet-lab experiments.
> >
> > Thank you for bringing up this point, and we appreciate your valuable feedback!
> >
> > ---
> > We greatly appreciate your insightful comments, as they will undoubtedly help us improve the quality of our article. If our response has successfully addressed your concerns and clarified the ambiguities, we respectfully hope that you consider raising the score of our article. Should you have any further questions or require additional clarification, we would be delighted to engage in further discussion. Once again, we sincerely appreciate your time and effort in reviewing our manuscript. Your feedback has been invaluable in improving our research.

---

> ### Author Response · Authors · 2023-08-18
> **Look forward to post-rebuttal feedback**
>
> Dear Reviewer SHsm,
>
> We would like to express our sincere gratitude for dedicating your time to reviewing our paper. Your insightful comments and suggestions have greatly contributed to enhancing the quality and clarity of our work.
>
> We have thoroughly considered your feedback and carefully responded to each of your questions. We would greatly appreciate your feedback on whether our responses have addressed your concerns to your satisfaction.
>
> Once again, we sincerely thank you for your invaluable contribution to our paper. As the rebuttal phase is progressing, we eagerly await your post-rebuttal feedback.
>
> Best regards,
> Authors.

---

> > ### Comment · Reviewer_SHsm · 2023-08-21
> > **Thank you for your comments**
> >
> > While I agree that the strong performance of non-deep models on larger datasets is indeed indicative that there is more than size of dataset impacting performance of deep models, and that it is plausible that the effects discussed in this paper are responsible, the key point that it is difficult to firmly and decisively prove this is the case: the argument that deep models fail on even large datasets is not a strong one in of itself; it is highly dependent on the structure and size of the deep model. Some deep models do indeed achieve good performance on OGB because of pretraining etc, but this is perhaps to be expected -- "designed" features such as molecular fingerprints need to be effectively learned in the pretraining phases of deep models. If the input dataset at pretraining time is either unrepresentative of the domain of molecules of interest, or too narrowly focused, the learned features in the earlier layers of the model are unlikely to prove suitable for the downstream task. As molecular fingerprints encode a great deal of human prior knowledge, they would be expected to perform better.
> >
> > Therefore, it is possible to argue that the poor performance of some deep models is simply due to poor training/ choice of pretraining data/ simply inadequate volumes of data to genuinely learn adequate feature extraction as compared to much larger image or text-focused models, especially when considering the very large and combinatorial nature of chemical space.
> >
> > However, I appreciate the effort the authors have put into addressing these concerns, and for bringing this discussion more sharply into focus for the research community. Given this I am raising my review score.
> >
> > Regarding the missing citations:
> > [1] Pappu et al. CoRR 2020 "Making graph neural networks worth it for low-data molecular machine
> > learning"
> > [2] Stanley et al. NeurIPS 2021 "FS-Mol: A Few-Shot Learning Dataset of Molecules"

---

> > > ### Author Response · Authors · 2023-08-21
> > > **Thanks for your post-rebuttal response**
> > >
> > > Dear reviewer SHsm,
> > >
> > > Thanks for your post-rebuttal response and insightful reviews. We promise to take your valuable advice into consideration and discuss the works you recommended in the future version. If you have any further questions or concerns, we remain dedicated to addressing them with the utmost eagerness.
> > >
> > > Best regards,
> > > Authors.

---

### Official Review · Reviewer_rZmN · 2023-07-06

**Soundness:** 3 good
**Presentation:** 3 good
**Contribution:** 2 fair
**Rating:** 4
**Confidence:** 4

**Summary:**

- The paper evaluates various (non-)deep models for molecular property prediction. Tree models using molecular fingerprints as inputs are shown to perform better than other competitors.
- The paper claims also to contain an extensive empirical investigation into the unique patterns of molecule data and inductive biases of various models underlying these phenomena.
- It also presents a feature mapping method for molecule data prior to feeding them into deep models.


**Strengths:**

- The topic certainly offers various interesting research directions and the paper mentions open questions.

- The study considers datasets beyond Moleculenet, namely, datasets related to COVID-19 and activity cliff datasets.

- The analysis in terms of smoothness is interesting, even if we know it. Basically, the paper shows that the molecular data often does not match this model bias of deep learning.

- The feature mapping is an interesting novelty but needs some clarification (see below).


**Weaknesses:**

- I do not fully understand why this paper was not submitted to the datasets & benchmark track, the research part seems minor (large parts are in the appendix), l67ff.:
We provide the most comprehensive benchmark on MPP tasks to date and expose the limitations of deep models on molecule datasets. Our findings offer new and valuable insights for the fast-growing AIDD community.


- There are important related works the paper is missing, which contain similar results, e.g.
  - Tom et al. Calibration and generalizability of probabilistic models on low-data chemical datasets with DIONYSUS, RSC Digital Discovery 2023
  - Deng et al. Taking a Respite from Representation Learning for
Molecular Property Prediction, arXiv preprint, arXiv:2209.13492, 2022

- Experimental Setting
  - The 2D baselines considered besides MPNN seem inappropriate to me. The study applies GCN and GAT, which do not even consider edge features. Models accepted by the community would more fitting, e.g., D-MPNN (Yang et al, 2019).
  - The study focuses on random splits while the scaffold split setting was suggested, e.g., with Moleculenet or the foundational study on GNN SSL by Hu et al. 2019, and is standard in all SOTA works.

- l. 174, Observation 2. It is irregular data patterns, NOT solely the small size of molecular datasets to blame for the failure of deep models! - as mentioned above, exactly because of this, the scaffold splitting has been accepted by the community. Either the authors missed that or I am misunderstanding the point.

- l. 183, Observation 3. Tree models (XGB and RF) exhibit a particular advantage over other models.
Given the above-mentioned works, I assume every researcher in the field runs some simple fingerprint-based baselines for comparison. Hence I do not think this is a novel finding for the community.

- To me, it's not clear in how far the feature mapping is better than the commonly used mapping to dense features (i.e., using torch.Embedding). Is the latter mapping used in the baselines? I would also suggest to move the theoretical justification from the appendix to the main text.



**Questions:**

Just the above-mentioned question about the feature mapping.

**Limitations:**

-

---

> ### Author Response · Authors · 2023-08-16
> **Response to Reviewer rZmN (1/3)**
>
> > *I do not fully understand why this paper was not submitted to the datasets & benchmark track, the research part seems minor (large parts are in the appendix), l67ff.: We provide the most comprehensive benchmark on MPP tasks to date and expose the limitations of deep models on molecule datasets. Our findings offer new and valuable insights for the fast-growing AIDD community.*
>
> Thanks for your valuable advice! It is worth noting that the datasets & benchmark track typically features papers that primarily introduce new datasets without novel approaches. Given that our work introduces a novel method to improve the performance of deep models, we believe that the main track is the appropriate venue to showcase our research.
>
> > *There are important related works the paper is missing.*
>
> Thank you for your recommendations! We appreciate your feedback and will discuss these important related works in detail in the future version of the paper. Specifically, Tom et al. [1] have conducted an extensive study on the calibration and generalizability of probabilistic machine learning models using small chemical datasets. Deng et al. [2] have performed a systematic evaluation of a collection of representative models using various molecular representations and have found that deep models still exhibit limited performance in molecular property prediction across most datasets.
>
> However, as we discussed in lines 42-46 of the paper, previous studies have not taken into account the emerging powerful deep models, such as SphereNet, nor have they explored the use of various molecular descriptors, such as 3D molecular graphs. Additionally, these studies have not thoroughly investigated the reasons behind the frequent failures of deep models in molecular applications.
>
> We acknowledge the importance of these related works and will ensure to include a comprehensive discussion and analysis of their findings in the revised version of the paper. Thank you for bringing them to our attention.
>
> > *The 2D baselines considered besides MPNN seem inappropriate to me. The study applies GCN and GAT, which do not even consider edge features. Models accepted by the community would more fitting, e.g., D-MPNN (Yang et al, 2019).*
>
> **Table Re1**:  The performance of D-MPNN and IFM-D-MPNN (D-MPNN equipped with our method IFM).
> | Models |BACE|HIV|BBBP| ClinTox| SIDER| Tox21| ToxCast| MUV| SARS-CoV-2|
> |----|----|----|----|----| ----  |---- |----| ----  |---- |
> |D-MPNN| 0.872| 0.823 | 0.897|0.896|0.619 |0.833 |0.772 | 0.050 |0.684 |
> | **IFM-D-MPNN**|0.897 |0.843 |0.936 |0.936| 0.642| 0.855| 0.801| 0.092| 0.703|
>
> Thanks for your valuable advice! In our experiments, the 2D baseline AttentiveFP (AFP)[3] also considers the edge features. Additionally, following your advice, we comprehensively evaluate D-MPNN and IFM-D-MPNN (D-MPNN equipped with our method IFM) on the molecule datasets. The results shown in Table Re1 indicate that our method can also improve the performance of D-MPNN.

---

> > ### Author Response · Authors · 2023-08-16
> > **Response to Reviewer rZmN (2/3)**
> >
> > > *The study focuses on random splits while the scaffold split setting was suggested, e.g., with Moleculenet or the foundational study on GNN SSL by Hu et al. 2019, and is standard in all SOTA works.  "It is irregular data patterns, NOT solely the small size of molecular datasets to blame for the failure of deep models! "- as mentioned above, exactly because of this, the scaffold splitting has been accepted by the community. Either the authors missed that or I am misunderstanding the point. *
> >
> > **Table Re2**:  The performance of various models on BACE (small) and HIV (large) with scaffold split.
> > | Models| SVM |XGB|RF| CNN| RNN| TRSF| MLP| GCN| MPNN| GAT| D-MPNN| AFP |
> > |----|----|----|----|----| ----  |---- |----|----|----| ----  |---- |---- |
> > | BACE (1,513) | 0.792 $\pm$ 0.014|0.803 $\pm$ 0.018 |0.826 $\pm$ 0.015|0.762 $\pm$ 0.023 |0.751 $\pm$ 0.036| 0.769 $\pm$ 0.025| 0.799 $\pm$ 0.027 | 0.781 $\pm$ 0.013| 0.786 $\pm$ 0.024|0.774 $\pm$ 0.022 |0.796 $\pm$ 0.014| 0.788 $\pm$ 0.025|
> > | HIV (93,087) | 0.782 $\pm$ 0.027| 0.796 $\pm$ 0.018| 0.785 $\pm$ 0.020| 0.743 $\pm$ 0.035|0.750 $\pm$ 0.025 |0.757 $\pm$ 0.019 | 0.776 $\pm$ 0.023| 0.761 $\pm$ 0.014|0.773 $\pm$ 0.024 | 0.762 $\pm$ 0.017|0.783 $\pm$ 0.020|0.772 $\pm$ 0.019|
> >
> > Thanks for your insightful review! Scaffold splitting has been adopted as a mean to evaluate the (out-of-distribution) generalization performance of machine learning models on novel chemical structures, i.e., the training and testing molecules contain distinct core scaffold structures. However, our study aims to verify that molecules with similar structures may have distinct properties, i.e., the training and testing molecules are expected to have similar structures. Therefore, we did not adopt scaffold splitting in our work.
> >
> > Additionally, Hu et al. (2019) adopted the scaffold split to evaluate the out-of-distribution prediction ability of the pre-trained models because they motivate their study by claiming the pre-trained models can transfer to out-of-distribution data. Also, they simplify the molecule featurization process, i.e., the input atom/bond features in their settings are not comprehensive (use a minimal set of node and bond features that unambiguously describe the 2D structure of molecules) because they did not focus on molecules, but instead to very the effectiveness of their methods in various graph data (molecule graphs, protein graphs, etc.). However, in chemoinformatics, many works [1,3,5] including the MoleculeNet paper [5] adopted the random split (for most datasets) and comprehensive atoms/bonds features to take full advantage of all the available information.
> >
> > Based on your valuable advice, we have also conducted evaluations of the 12 models on two datasets using scaffold splitting. The results presented in Table Re2 demonstrate that non-deep models can outperform most deep models, both on small-scale (BACE) and large-scale (HIV) datasets. These findings align with our conclusion that there is an additional crucial factor contributing to the limitations of deep models, which goes beyond the small size of the molecule datasets.
> >
> > > *Tree models (XGB and RF) exhibit a particular advantage over other models. I do not think this is a novel finding for the community.*
> >
> > Thanks for your careful reviews! Firstly, the main contribution of our study lies in the explanations of these observations and the proposed methods to improve the performance of deep models. We aim to provide a comprehensive understanding of the challenges faced by deep models in molecular property prediction. Secondly, regarding the special superiority of tree models (conclusion 3), we are not aware of previous works that have specifically highlighted this observation in molecular property prediction. If you could kindly point us to any relevant studies, we would greatly appreciate it. Furthermore, many previous studies simply feed one fingerprint into tree models. However, with the concatenation of several readily available fingerprints as input, the superiority of tree models would be more significant. Thank you once again for your valuable feedback!

---

> > > ### Author Response · Authors · 2023-08-16
> > > **Response to Reviewer rZmN (3/3)**
> > >
> > > > *To me, it's not clear in how far the feature mapping is better than the commonly used mapping to dense features (i.e., using torch.Embedding). Is the latter mapping used in the baselines?*
> > >
> > > **Table Re3**:  Comparisons between IFM and *torch.Embedding*. **BACE, HV, BBBP: Classification; ESOL, Lipop, FreeSolv: Regression.**
> > > | Models |BACE|HIV|BBBP| ESOL| Lipop| FreeSolv|
> > > |----|----|----|----|----| ----  |---- |
> > > | MLP | 0.887| 0.791 |0.918 | 0.653|0.633 | 1.046|
> > > | MLP + *torch.Embedding* | 0.885| 0.802 |0.916 | 0.662| 0.637| 1.004|
> > > | **IFM-MLP (Ours)**| 0.915|0.816  |0.937 | 0.587| 0.556| 0.862|
> > > | GCN|0.880 |0.834  |0.915 | 0.653| 0.633| 1.046|
> > > | GCN + *torch.Embedding*| 0.883| 0.829 | 0.917|0.647 |0.628 |1.062 |
> > > | **IFM-GCN (Ours)**| 0.903|0.862  |0.945 |0.728 |0.577 |0.916 |
> > >
> > > Thanks for the helpful reviews! We use *torch.Embedding* to map the input the atoms/bonds features to real vectors as the previous work [4]. The results shown in Table Re3 indicate that IFM (our method) can consistently beat the simple torch.Embedding. As shown in the theoretical justification (in the appendix), IFM can help the models fit the un-smooth target functions while *torch.Embedding* does not possess such ability.
> > >
> > >
> > > > *I would also suggest to move the theoretical justification from the appendix to the main text.*
> > >
> > > Thanks for your helpful advice! We would move the theoretical justification from the appendix to the main text in the future version following your valuable advice.
> > >
> > > [1] Calibration and generalizability of probabilistic models on low-data chemical datasets with DIONYSUS, RSC Digital Discovery 2023
> > > [2] Taking a Respite from Representation Learning for Molecular Property Prediction, arXiv preprint, arXiv:2209.13492, 2022
> > > [3] Pushing the boundaries of molecular representation for drug discovery with graph attention mechanism, J Med Chem, 2020
> > > [4] Strategies for Pre-training Graph Neural Networks, ICLR, 2020.
> > > [5] MoleculeNet: a benchmark for molecular machine learning, Chemical Science, 2017.
> > >
> > > ---
> > > We greatly appreciate your insightful comments, as they will undoubtedly help us improve the quality of our article. If our response has successfully addressed your concerns and clarified the ambiguities, we respectfully hope that you consider raising the score of our article. Should you have any further questions or require additional clarification, we would be delighted to engage in further discussion. Once again, we sincerely appreciate your time and effort in reviewing our manuscript. Your feedback has been invaluable in improving our research.

---

> ### Comment · Reviewer_rZmN · 2023-08-17
> **Thank you for the detailed responses!**
>
> I think the contribution is decent and could get interesting in terms of in presenting and analyzing the feature mapping. However, the this would greatly change the present paper and should probably get a second reviewing. In particular, in view of the overall critical initial reviews.

---

> > ### Author Response · Authors · 2023-08-18
> > **The rebuttal process prevents misunderstandings or critical initial reviews from leading to outright rejection.**
> >
> > Dear reviewer rZmN,
> >
> > Thank you for your constructive feedback on our manuscript, particularly regarding the recommended inclusion of more related works and conducting comparisons between IFM and other embeddings. We appreciate your recognition of the potential interest and value of our contribution.
> >
> >  **We believe that the rebuttal and author-reviewer discussions play a crucial role in preventing misunderstandings or critical initial reviews from leading to outright rejection of the manuscript. If our responses adequately address the concerns you have raised, we respectfully hope that you can increase your score to support our work.**
> >
> > We greatly appreciate the time and effort you have dedicated to reviewing our work. If you have any further questions or concerns, we remain dedicated to addressing them with the utmost eagerness.
> >
> > Best regards,
> > Authors.

---

> ### Comment · Reviewer_rZmN · 2023-08-18
> **Final Response**
>
> I am sorry but I (am one of the few who actually) took part in the internal reviewer discussion and agree with the (current) conclusions obtained there.
>
> Borderline accept means "Technically solid paper where reasons to accept outweigh reasons to reject, e.g., limited evaluation." This does not reflect my opinion of the current paper, including the clarifications about the method in the rebuttal. Of course, the focus and nature of the paper could change, but I think this would be a rather different paper then. If you consider my reviewing, conclusions and overall slightly negative rating as outright rejection then we probably just disagree. My rating can easily be overruled by the remaining reviewers and I will continue following the discussion.

---

> > ### Author Response · Authors · 2023-08-21
> > **Thanks for your response**
> >
> > Dear reviewer rZmN,
> >
> > Thanks for your post-rebuttal response and insightful reviews. We promise to take your valuable advice into consideration in the future version. If you have any further questions or concerns, we remain dedicated to addressing them with the utmost eagerness.
> >
> > Best regards,
> > Authors.

---

### Official Review · Reviewer_ah6B · 2023-07-06

**Soundness:** 1 poor
**Presentation:** 2 fair
**Contribution:** 2 fair
**Rating:** 3
**Confidence:** 4

**Summary:**

This paper investigates why classical ML models (i.e. anything except neural networks) often outperform neural networks on molecular property prediction tasks.

The paper first presents a large study evaluating many deep and non-deep methods on a variety of datasets (seems to be mostly MoleculeNet but I was unsure of this detail). The conclusions are:

1. Classical models do tend to perform better
2. Dataset size is not the only factor responsible for low performance of deep methods
3. Tree methods (e.g. random forest + XGBoost) do particularly well

The authors speculate that this is due to molecular functions not being smooth and mixing of input features. They do some experiments which provide some evidence for this (although I don't think they are conclusive, more on this below).

The paper then proposes to fix this by transforming the input features with a sinusoidal transformation, and show experimentally that this improves performance.

**EDIT**: I read the rebuttal and it improved my opinion of the paper, but only slightly. I still believe the significance/novelty of the findings is not super high.

**Strengths:**

I think the biggest strength of this paper is the problem choice: not many works study this question and I think it is arguably *the* most important question in research on ML for molecules.

I also think this paper has some good ideas for studying this problem: e.g:
- looking at performance on activity cliff vs non-activity cliff molecules.
- doing experiments on an artificially smoothed version of the dataset
- performing experiments with artificially mixed input features

Unfortunately I don't think the experiments fully support the authors' conclusions, but I will discuss that further in weaknesses.

The idea of feature transformation is interesting but not very well-explained.

**Originality / quality / clarity / significance**

My explicit opinion on these aspects of the paper (also accounting for the weaknesses in the next section) are:

- *Originality*: although this is the only paper I know of which specifically addresses deep vs non-deep models for molecules, this question has been previously studied in more general settings (e.g. https://arxiv.org/abs/2108.13637) and countless papers in cheminformatics have compared NNs and classical methods. The idea of studying activity cliffs was studied of course in the paper introducing the activity cliff dataset. Therefore I think the originality of the paper is not super high, although there is some originality.
- *Quality*: I think the key analysis of the paper is not very thorough, so I must unfortunately rate this as medium/low.
- *Clarity*: I thought this was ok, but the paper was made less clear because many important details were relegated to the appendix without any indication of where this information could be found.
- *Significance*: I could imagine some ideas from this paper being referenced in future work, but I think the conclusions are not fully supported by the experiments (more next section) so I think the significance of the paper is fairly limited.

**Weaknesses:**

There are a few weaknesses I would like to point out in this paper, which I will list roughly in order of importance:

**Experiments don't seem to support conclusions**

Specifically:
- In section 3, the authors conclude that "irregular data patterns, NOT solely the small size of molecular datasets [are] to blame for the failure of deep models". However, "irregular data patterns" is not really defined: what makes a molecule pattern "irregular"? This seemed to just be giving a name to whatever phenomenon causes dataset size to not totally predict performance. The other 2 conclusions from this section are supported but just confirm what is in numerous previous works.
- In section 4, "smoothness" is not defined precisely. Although deep models perform better than non-deep models in Figure 2, the error for all methods decreases sharply, and the differences between all methods are small compared to the overall change in performance, so the statistical significance of this result is questionable. My conclusion would just be that the smoothed tasks are much easier! Also, the number of datasets is very limited. Perhaps the authors could extend this to the classification dataset by using a mixture of labels, e.g. like MIXUP (https://arxiv.org/abs/1710.09412)
- Similarly, in section 4 the authors conclude from their experiments on "activity cliff" and "non-activity cliff" molecules that deep models are "less sensitive to subtle structural changes and struggle to learn non-smooth target functions" because the performance difference between activity cliff and non-activity cliff molecules is small. I think this cannot be concluded by looking at error rates. You would need to look at how the predictions change as the input structure is perturbed. The overall error rate is high for both cliff and non-cliff molecules, so I would instead conclude that deep model just fit poorly to both the cliff and non-cliff datasets, and that smoothness is *not* affecting the quality of the fit.
- At the end of section 4 the authors conclude that molecular features should not be mixed, even though this does not affect the performance of deep models. I agree that one would expect this to worsen the performance of tree models, but as the authors themselves note a deep model could learn the inverse transformation and therefore be less affected. It seemed like the authors concluded the opposite of what I expected?
- Experiments with the sinusoidal featurization in section 6 are not comprehensive: the authors don't seem to control for many other factors including regularization, early stopping, etc. There are too many factors going into neural network training to conclude that their featurization is effective from such a simple study.
- The tables do not contain error bars, so the statistical significance of performance differences are unclear

**Clarity**

The paper was not very clear, with many details omitted or put into unspecified areas of the appendix. Some key thing which were unclear are:

- The fingerprint featurization in section 3.1: how many dimensions were dropped? Are the vectors no longer sparse?
- Neural network training: how was regularization handled? Was the number of parameters varied to avoid overfitting?
- The featurization in section 5 is barely described...

**Related work**

The authors did not comment on numerous other works which compare deep and non-deep methods in some capacity, when this is arguably the most important related work. The related work section only introduced the descriptors.

**Questions:**

See weaknesses section. I would mainly like the authors to respond to my questions about whether their experiments support their conclusions.

**Limitations:**

The main limitation of the study is that the conclusions are not well-supported in my opinion.

---

> ### Author Response · Authors · 2023-08-16
> **Response to Reviewer ah6B (1/3)**
>
> > *In section 3,  "irregular data patterns" is not really defined: what makes a molecule pattern "irregular"? The other 2 conclusions from this section are supported but just confirm what is in numerous previous works.*
>
> In our study, “irregular data patterns” refer to the phenomenon where small structural variations in molecules can lead to significant changes in their properties. This means that the mapping function from molecules to properties is unsmooth, and even slight changes in the input can result in drastic variations in the model’s predictions. In contrast, in image data, changing a few pixels usually does not completely alter the image category. We apologize for not providing a clear definition of “irregular data patterns” in Section 3, but we do explain this concept in more detail in Section 4. Based on your valuable advice, we will make sure to define “irregular data patterns” clearly in Section 3 in future versions of the manuscript.
>
> Furthermore, the main contribution of our study lies in the explanations of these observations and the proposed methods to improve the performance of deep models. We aim to provide a comprehensive understanding of the challenges faced by deep models in molecular property prediction. Regarding the special superiority of tree models (conclusion 3), we are not aware of previous works that have specifically highlighted this observation in molecular property prediction. If you could kindly point us to any relevant studies, we would greatly appreciate it.
>
> Thank you once again for your valuable feedback. We will make the necessary revisions and improvements to enhance the clarity and contributions of our manuscript.
>
> > *In section 4, "smoothness" is not defined precisely. Although deep models perform better than non-deep models in Figure 2, ..., my conclusion would just be that the smoothed tasks are much easier!
>
> Thanks for your careful reviews! We define the target functions mapping molecules to properties as “unsmooth” because small structural variations in molecules, such as replacing an atom or bond, can result in significant changes in their properties.
>
> Additionally, kindly note that our objective is not to establish that the smoothed tasks are inherently easier, although it is indeed true. Instead, our aim is to demonstrate that deep models are better suited for the molecule datasets with higher smoothing levels. Specifically, when the data is not smoothed (0-smooth), non-deep models generally perform better. As we increase the smoothing level (10-smooth and 20-smooth) of the molecule data, deep models gradually outperform non-deep models in general. We have provided a detailed analysis of this phenomenon in lines 202-206 of our manuscript.
>
> The focus of our study is the comparison between non-deep models and deep models under different smoothing levels, rather than the overall performance change of different models. Thanks for your contribution to improving the clarity of our work.
>
> > *The number of datasets (for smoothing experiments) is limited. Perhaps the authors could extend this to the classification dataset by using a mixture of labels, e.g.  MIXUP.*
>
> **Table Re1**: The performance of various models on the smoothed FreeSolv dataset.
> | Models| SVM |XGB|RF| MLP| GCN| MPNN| GAT| AFP |
> |----|----|----|----|----| ----  |---- |----|----|
> | 0-smooth | 1.063|0.715|1.014|1.046 |1.316 |1.327|1.304|1.139 |
> | 10-smooth | 0.842|0.827|0.840|0.751 |0.814 |0.931|0.818|0.952 |
> | 20-smooth|0.683 |0.685|0.693|0.623 |0.653 |0.701|0.665|0.681 |
>
> Thank you for your valuable advice! We appreciate your suggestion to extend the smoothing experiments to the classification dataset using a mixture of labels, such as MIXUP. However, there are several reasons why we did not apply MIXUP to the classification datasets in our study.
>
> Firstly, MIXUP generates label mixtures that are no longer in a one-hot or binary format. This poses challenges in calculating evaluation metrics such as testing accuracy or AUC-ROC for classification tasks.
>
> Secondly, some of the non-deep models we compared in our study, such as Random Forests and SVMs, typically do not use cross-entropy as the loss function. Therefore, the smoothing experiments using MIXUP are not directly applicable to these classification models.
>
> Furthermore, applying MIXUP to molecular graphs can be extremely complex due to the need to align the nodes of two molecules before mixing them [1]. This alignment process introduces additional complexity and may affect the effectiveness of applying MIXUP directly to molecular graph classification.
>
> To address your concern, we have expanded our experiments to include another regression dataset, FreeSolv. The results are reported in Table Re1. As observed, when the data is not smoothed (0-smooth), non-deep models generally perform better. However, as we increase the smoothing level (10-smooth and 20-smooth) of the molecule data, deep models gradually outperform non-deep models in general.

---

> > ### Author Response · Authors · 2023-08-16
> > **Response to Reviewer ah6B (2/3)**
> >
> > > *In section 4 the authors conclude from their experiments on "activity cliff" and "non-activity cliff" molecules that deep models are "less sensitive to subtle structural changes and struggle to learn non-smooth target functions" because the performance difference between activity cliff and non-activity cliff molecules is small. I think this cannot be concluded by looking at error rates. You would need to look at how the predictions change as the input structure is perturbed. The overall error rate is high for both cliff and non-cliff molecules, so I would instead conclude that deep model just fit poorly to both the cliff and non-cliff datasets, and that smoothness is not affecting the quality of the fit.*
> >
> > **Table Re2**: The predictions change rates of various models when transitioning from non-cliff to cliff molecules.
> > | Models| SVM |XGB|RF| CNN| RNN| TRSF| MLP| GCN| MPNN| GAT| D-MPNN| AFP |
> > |----|----|----|----|----| ----  |---- |----|----|----| ----  |---- |---- |
> > | CB1 | 15.04%| 21.13%|20.76% | 2.07%| 10.15%|9.42% |13.32%|4.13% |3.85% |1.17% |4.01%|4.35%|
> > | DAT |20.64% | 23.03% |23.95% |2.73%| 14.18%|9.34%|15.02%|5.83% |5.15% | 2.27%|4.33%|5.08%|
> > | PPAR$\alpha$| 21.07%| 22.93%|23.14% | 11.29%| 13.48%|18.39% |15.29% |1.83% |5.18% |4.93%|4.85%|11.73%|
> > | DOR|25.26%| 28.41%| 23.95% | 10.02%| 9.83%| 10.25%| 15.18%|9.77% |12.52% |11.36% |12.09%|13.11%|
> >
> > Thanks for your insightful reviews! We appreciate your valuable input. Based on your suggestion, we have included the prediction change rates of various models when transitioning from non-cliff to cliff molecules with subtle structural changes. As shown in Table Re2, the deep models exhibit less significant prediction change rates compared to the non-deep models. This observation suggests that deep models are indeed less sensitive to subtle structural changes compared to non-deep models.
> >
> > Regarding the impact of smoothness on the quality of fit, we respectfully disagree with your conclusion. In Table 2 of our manuscript, we demonstrate that the prediction RMSE for cliff molecules is larger than that for non-cliff molecules. This difference indicates that the deep models struggle to accurately capture the non-smooth target functions associated with cliff molecules, resulting in poorer fit quality.
> >
> > We sincerely appreciate your helpful advice and feedback, which have contributed to the improvement of our manuscript.
> >
> > > *At the end of section 4 the authors conclude that molecular features should not be mixed, even though this does not affect the performance of deep models. I agree that one would expect this to worsen the performance of tree models, but as the authors themselves note a deep model could learn the inverse transformation and therefore be less affected. It seemed like the authors concluded the opposite of what I expected?*
> >
> > Thank you for your thoughtful response! We would like to clarify that our claim is that "molecular features should not be combined prior to inputting them into models" (Lines 261-262). Here, the term "models" refers specifically to non-deep learning models such as SVM, RF, and XGB. We apologize for any ambiguity in our description and assure you that we will address this in future versions. Once again, we sincerely appreciate your valuable feedback!
> >
> >
> > > *Experiments with the sinusoidal featurization in section 6 are not comprehensive: the authors don't seem to control for many other factors including regularization, early stopping, etc. There are too many factors going into neural network training to conclude that their featurization is effective from such a simple study.*
> >
> > Thanks for your helpful reviews! We would like to clarify that in our study, we ensured that the other settings, including regularization, early stopping, and hyperparameters, were kept the same as the benchmarking experiments. We mentioned this in Line 293 of our manuscript. The specific experimental settings of various models in the benchmarking experiments can be found in our manuscript (Section3 of the main text and Section 11 of the appendix).
> >
> > > *The tables do not contain error bars, so the statistical significance of performance differences are unclear.*
> >
> > Thanks for your careful reviews. We would like to clarify that the standard deviations corresponding to the performance measurements can be found in the appendix (Section 7) of our manuscript due to space limitations of the main text.
> >
> > [1] Graph Mixup with Soft Alignments (ICML 2023)

---

> > > ### Author Response · Authors · 2023-08-16
> > > **Response to Reviewer ah6B (3/3)**
> > >
> > > > *The fingerprint featurization in section 3.1: how many dimensions were dropped?*
> > >
> > > Thank you for your careful review. Regarding the specific number of dimensions that were dropped, we did not provide this information because the number can vary across different molecular datasets and is dependent on the pre-processing procedures (rules) applied (Line 123 - 126): (1) remove features with missing values, (2) remove features with extremely low variance (variance < 0.05), and (3) remove features that had a high correlation (Pearson correlation coefficient > 0.95) with another feature. For example, for Sider dataset, we drop 1038 dimensions with above rules.
> > >
> > > > *Neural network training: how was regularization handled? Was the number of parameters varied to avoid overfitting?*
> > >
> > > Thanks for your valuable feedback! We use the $L_2$ regularization in our experiments and the tuning space of hyperparameter $\lambda$ can be found in section 11 of the appendix. Additionally, we adopted dropout to avoid overfitting and the dropout rate can also be found in section 11 of the appendix.
> > >
> > > > *The featurization in section 5 (Methodology) is barely described...*
> > >
> > > Thanks for your careful reviews. We have mentioned that the other settings are the same as the benchmarking experiments in Line 293. Namely, the fingerprint featurization is the same as the pre-processing procedures in section 3.1. And then, we conduct the Independent Feature Mapping (our method) on the features after these pre-processing procedures.
> > >
> > > ---
> > > We greatly appreciate your insightful comments, as they will undoubtedly help us improve the quality of our article. If our response has successfully addressed your concerns and clarified any ambiguities, we respectfully hope that you consider raising the score of our article. Should you have any further questions or require additional clarification, we would be delighted to engage in further discussion. Once again, we sincerely appreciate your time and effort in reviewing our manuscript. Your feedback has been invaluable in improving our research.

---

> > > > ### Comment · Reviewer_ah6B · 2023-08-16
> > > > **Response to rebuttal**
> > > >
> > > > Thanks for reading my review and carefully answering my questions. Your response has improved my impression of the paper so I raised my score a bit (but still not recommending acceptance). I'll make select comments in my response:
> > > >
> > > > > Regarding the special superiority of tree models (conclusion 3), we are not aware of previous works that have specifically highlighted this observation in molecular property prediction. If you could kindly point us to any relevant studies, we would greatly appreciate it.
> > > >
> > > > ### Superiority of RFs
> > > >
> > > > I think this is generally well-known in the field: ask anybody in chemistry, Two studies I can think of are:
> > > >
> > > > 1. Stanley, Megan, et al. "Fs-mol: A few-shot learning dataset of molecules." Thirty-fifth Conference on Neural Information Processing Systems Datasets and Benchmarks Track (Round 2). 2021.
> > > > 2. Calibration and generalizability of probabilistic models on low-data chemical datasets with DIONYSUS
> > > >
> > > > In fact, study 2 was also pointed out by another reviewer. I've seen countless other studies report higher scores by tree-based methods, but I don't have time to go back and look for them, sorry.
> > > >
> > > > ### Smoothing/cliff experiments
> > > >
> > > > In section 3-4 you performed experiments on a series of tasks (e.g. cliff/non-cliff, varying levels of smoothness) where the overall trend was that the tasks are easier, but you made conclusions based on the *relative* trends. After reading your response and thinking a bit more about it, I think this is a valid experimental method, but I still think you need to be careful with drawing conclusions from this kind of study where multiple things change at once. There could be other things causing the performance difference: for example, differences in optimal hyperparameters (I understood that you hold hyperparameters constant), effective removal of outliers. Therefore I think it is not clear that on a "natural" smooth dataset the same trends would hold, although I think your study does provide some evidence.
> > > >
> > > > ### Clarifications
> > > >
> > > > Thank you for clarifying some of your claims and training details. I generally agree with those choices/conclusions.
> > > >
> > > > Overall my biggest remaining concerns are novelty of the conclusions and the correctness of the IFM method. I will discuss this with the other reviewers.

---

> > > > > ### Author Response · Authors · 2023-08-17
> > > > > **Thanks for your response!**
> > > > >
> > > > > Thank you for your prompt response!
> > > > >
> > > > > >* **Q1:** The observation of the superiority of RFs is not novel.*
> > > > >
> > > > > **A1:** Thanks for your valuable feedback!  **Kindly that the main contribution of our study does not lie in the three observations listed in section 3, but instead in the explanations of these observations and the proposed methods to improve the performance of deep models.**  Actually, we have also mentioned some works [1, 2, 3] that claimed that deep models struggled to outperform non-deep ones in our manuscript (Line 42 - 46).  However,  they neither explore the reasons why deep models would often fail nor provide any solution accordingly.
> > > > >
> > > > > Additionally, most previous works claimed that **non-deep models** would often perform better. In contrast, **we highlight the special superiority of **tree models** (RFs and XGB) instead of the general non-deep models, which to the best of our knowledge has not been exposed by previous works in chemistry**.
> > > > >
> > > > > >* **Q2:** Smoothing/cliff experiments are valid. But there could be other things causing the performance difference: for example, differences in optimal hyperparameters (I understood that you hold hyperparameters constant), and effective removal of outliers. Therefore I think it is not clear that on a "natural" smooth dataset the same trends would hold, although I think your study does provide some evidence.*
> > > > >
> > > > > **A2:**  We appreciate that you think our experiments are valid to verify the conclusions.  However, we respectfully disagree with your conclusion that there could be other things causing the performance difference and biased conclusions. The reasons are as follows:
> > > > >
> > > > > Firstly, we did not adopt the same hyperparameters between the datasets and their smoothed version. Instead, we use the same hyperparameter tunning space and select the optimal hyperparameter set using the same TPE algorithm (Line 149 -151). In addition,  we evaluated the same well-trained models on both non-cliff and cliff testing molecules and observe significant performance difference. The testing process did not involve hyperparameters tunning.  Therefore, the hyperparameters would not cause the performance difference.
> > > > >
> > > > > Secondly, there are no outliers in the smoothed version of the "normal" datasets because we only change the molecules' labels. Also, we do not need to remove any outliers in the activity cliff datasets. Therefore, the outliers would not cause the performance difference.
> > > > >
> > > > > Additionally, although we smooth the "normal" datasets artificially, the activity cliff datasets in our experiments are **natural** because they come from real-world practice in the AI-assisted Drug Discovery (AIDD)  tasks. Therefore, we think that even if on "natural" datasets the same trends would hold.
> > > > >
> > > > > [1]  Analyzing learned molecular representations for property prediction. (JCIM 2019)
> > > > > [2] Large-scale comparison of machine learning methods for drug prediction on chembl. (Chemical Science 2018)
> > > > > [3] Predicting molecular activity on nuclear receptors by multitask neural networks. (Journal of Chemometrics 2022)
> > > > >
> > > > > ---
> > > > >
> > > > > Thanks again for your valuable advice in terms of the clarity and the number of smoothing datasets, **especially the experiments of predictions change rates you suggested us to do**.  We promise to carefully revise our manuscript based on your advice above. Also, if you have any further questions or concerns, we remain dedicated to addressing them with the utmost eagerness. Once again, we greatly appreciate your insightful comments, as they will undoubtedly help us improve the quality of our article.

---

> > > > > > ### Author Response · Authors · 2023-08-19
> > > > > > **Have our response addressed your concerns?**
> > > > > >
> > > > > > Dear reviewer ah6B,
> > > > > >
> > > > > > We greatly appreciate the time and effort you have dedicated to reviewing our work. We have carefully responded to your questions/concerns and have conducted extensive experiments to address them.
> > > > > >
> > > > > > As the discussion deadline is approaching, we want to check if our response has adequately addressed your concerns. If you have any further questions or concerns, we remain dedicated to addressing them with the utmost eagerness.
> > > > > >
> > > > > > Once again, we sincerely appreciate your valuable input, and we look forward to hearing from you soon.
> > > > > >
> > > > > > Best regards,
> > > > > > Authors.

---

> > > > > > > ### Author Response · Authors · 2023-08-21
> > > > > > > **Look forward to post-rebuttal feedback**
> > > > > > >
> > > > > > > Dear reviewer ah6B,
> > > > > > >
> > > > > > > Thanks for your valuable advice in terms of the clarity and the number of smoothing datasets, especially the experiments of prediction change rates you suggested we do. Given the fact that you keep your rating as ‘reject’, we want to know which specific concerns you still have. We remain dedicated to addressing them with the utmost eagerness. Look forward to further discussions!
> > > > > > >
> > > > > > > Best,
> > > > > > > Authors.

---

### Official Review · Reviewer_1yt1 · 2023-07-07

**Soundness:** 3 good
**Presentation:** 3 good
**Contribution:** 4 excellent
**Rating:** 7
**Confidence:** 4

**Summary:**

The authors demonstrate, through a large number of experiments across many datasets and tasks, that tree-based models on molecular fingerprints consistently outperform deep learning models in molecular property prediction, and attribute this to 1) the "non-smooth" nature of molecular labels (e.g. the existence of "activity cliffs"), and 2) the manner in which deep learning models mix molecular features. Given these conclusions, the authors propose a novel molecular feature pre-processing method called Independent Feature Mapping (IFM) and show that using IFM-transformed features as inputs to deep learning models allows them to outperform tree-based models.

**Strengths:**

### Originality

The authors are certainly not the first to point out that deep learning methods are often outperformed by non-deep models, especially tree-based models. However, the non-trivial explanation for this phenomenon in terms of the "non-smooth", high-frequency nature of molecular labels and the spectral bias of deep learning methods, as well as the introduction of IFM to deal with these issues, are important contributions to the understanding of how to improve molecular property prediction generally.

### Quality

The quality of the paper is high, containing a large number of relevant experiments across various datasets, tasks, and models. The appendix suggests that a suitable amount of hyperparameter tuning was performed for each model.

### Clarity

The paper is mostly clear, though a number of improvements can be made (see Weaknesses)

### Significance

Accurate OOD molecular property prediction is an incredibly valuable tool for drug discovery and material design, and this paper makes an important contribution to this area of research.

**Weaknesses:**

- A number of the tables and figures could be significantly improved to make them more easily comprehended. For example:
  - the lack of standard deviations in Table 1 makes the conclusions less significant, please move from appendix to main text.
  - Figures 2 and 4 should be summarized as tables and only the percent difference from 0-smooth or Original, respectively, displayed
  - Tables 5 and 6 could be moved to the appendix to make room.
- The caption of Figure 2 claims that the smoothing experiments were only done for regression tasks because smoothing is unsuitable for classification, but this is not true - cross-entropy loss can be applied to both one-hot and dense labels. Please provide a classification example.
- In Explanation 2, the following claim is made without evidence: "The learning style of tree models is more suitable for molecular data because only a handful of features (e.g., certain substructures) are most indicative of molecular properties." Please provide evidence for this claim.
- In Section 6.2, the following statement is made: "mixing different dimensions of molecular features as SM [33] and GM [48] would degrade the performance." However, only GM mixes features. Please fix.
- The Discussion section is short and weak. Possible limitations of the proposed approach are not discussed. Please update.

**Questions:**

- Why is batch size not treated as a hyperparameter? Section 3.1 suggests it is kept fixed at 128.
- The molecules in Figure 3 would have very similar structural fingerprints, so why would tree-based methods on these fingerprints do well on this example? Is there a better example that could illustrate the point being made?

**Limitations:**

No mention of limitations is made.

---

> ### Author Response · Authors · 2023-08-15
> **Response to Reviewer 1yt1 (1/2)**
>
> > *Some tables and figures could be significantly improved to make them more easily comprehended.*
>
> Thank you for your valuable advice, which will undoubtedly improve the quality and clarity of our manuscript! We promise to refine the tables and figures following your advice.
>
> > *The caption of Figure 2 claims that the smoothing experiments were only done for regression tasks because smoothing is unsuitable for classification, but this is not true - cross-entropy loss can be applied to both one-hot and dense labels. Please provide a classification example.*
>
> Thanks for your constructive reviews! Kindly note that we smooth the labels of both the training and testing datasets in our experiments. However, the reason we state that smoothing experiments are not suitable for classification tasks is twofold. First, after the smoothing, the labels are no longer in a one-hot or binary format, which makes it challenging to calculate testing accuracy or AUC-ROC for classification evaluation. Second, the models we compare in our study, namely Random Forests and SVMs, typically do not use cross-entropy as the loss function. Hence, the smoothing experiments are not directly applicable to these classification models.
>
> > *"The learning style of tree models is more suitable for molecular data because only a handful of features (e.g., certain substructures) are most indicative of molecular properties." Please provide evidence for this claim.*
>
> Thanks for your helpful reviews! Fragment-Based Drug Discovery (FBDD) has emerged as a drug discovery approach because molecule fragments (also known as functional groups) are most indicative of molecular properties [1]. Each dimension in the molecular fingerprints usually corresponds to a specific molecular fragment [2]. Therefore, the tree models that make decisions based on each dimension would capture the meaningful fragments more easily.
>
>
> > *In Section 6.2, the following statement is made: "mixing different dimensions of molecular features as SM  and GM would degrade the performance." However, only GM mixes features. Please fix.*
>
> Thanks for your careful reviews! We would fix this issue following your valuable advice.
>
> > *The molecules in Figure 3 would have very similar structural fingerprints, so why would tree-based methods on these fingerprints do well on this example? Is there a better example that could illustrate the point being made?*
>
> **Table Re1**: The predictions change rates of various models when transitioning from non-cliff to cliff molecules.
> | Models| SVM |XGB|RF| CNN| RNN| TRSF| MLP| GCN| MPNN| GAT| D-MPNN| AFP |
> |----|----|----|----|----| ----  |---- |----|----|----| ----  |---- |---- |
> | CB1 | 15.04%| 21.13%|20.76% | 2.07%| 10.15%|9.42% |13.32%|4.13% |3.85% |1.17% |4.01%|4.35%|
> | DAT |20.64% | 23.03% |23.95% |2.73%| 14.18%|9.34%|15.02%|5.83% |5.15% | 2.27%|4.33%|5.08%|
> | PPAR$\alpha$| 21.07%| 22.93%|23.14% | 11.29%| 13.48%|18.39% |15.29% |1.83% |5.18% |4.93%|4.85%|11.73%|
> | DOR|25.26%| 28.41%| 23.95% | 10.02%| 9.83%| 10.25%| 15.18%|9.77% |12.52% |11.36% |12.09%|13.11%|
>
> Thanks for your insightful reviews! Kindly note that Figure 3 is an example of activity cliffs, which means a situation where small changes in the chemical structure of a drug lead to significant changes in its property. In such cases, the target function mapping molecules to bioactivity is extremely un-smooth, meaning that small structural changes can result in drastic variations in molecular properties.
>
> The tree models are more suitable for fitting the un-smooth target functions because their decision boundaries are defined by the thresholds set on individual features. Tree models that make decisions based on each dimension (usually corresponding to a specific substructure) would capture subtle structural changes more easily.
>
> We also verify this point with extensive experiments in Table Re1. Specifically, we report the prediction change rates of various models when transitioning from non-cliff to cliff molecules, where subtle structural changes result in significant variations in molecular properties. As can be observed in Table Re1, the prediction change rates of deep models are less significant than tree models, indicating that tree models are indeed more sensitive to subtle structural changes. In contrast, deep models are better at fitting smooth functions, which is known as spectral bias [3] in deep learning theory.
>
> [1] The rise of fragment-based drug discovery (Nature Chemistry 2009)
> [2] Extended-Connectivity Fingerprints (JCIM 2020)
> [3] On the spectral bias of neural networks. (ICML 2019)

---

> > ### Author Response · Authors · 2023-08-15
> > **Response to Reviewer 1yt1 (2/2)**
> >
> > > *The Discussion section is short and weak. Possible limitations of the proposed approach are not discussed. Please update.*
> >
> > Thanks for your helpful reviews! We have discussed the limitations of our approach in Section 4 of the appendix (for the limited space). More specifically, IFM unavoidably incurs more computational overhead compared with the vanilla models because it would increase the dimensions of the molecular features. We note that a promising avenue for further research would be to find less computationally costly ways to improve deep models’ abilities in learning non-smooth target functions.
> >
> > > *Why is batch size not treated as a hyperparameter? Section 3.1 suggests it is kept fixed at 128.*
> >
> > Thanks for your helpful advice! We set the batch size as 128 for all the models for fairness. While batch size can have an impact on model training, it is often considered as a practical parameter that is determined by hardware limitations and computational considerations rather than to be optimized.
> >
> > ---
> > We greatly appreciate your insightful comments, as they will undoubtedly help us improve the quality of our article. If our response has successfully addressed your concerns and clarified any ambiguities, we respectfully request that you consider raising the score of our article. Should you have any further questions or require additional clarification, we would be delighted to engage in further discussion. Once again, we sincerely appreciate your time and effort in reviewing our manuscript. Your feedback has been invaluable in improving our research.

---

> ### Author Response · Authors · 2023-08-18
> **Look forward to post-rebuttal feedback**
>
> Dear Reviewer 1yt1,
>
> We would like to express our sincere gratitude for dedicating your time to reviewing our paper. Your insightful comments and suggestions have greatly contributed to enhancing the quality and clarity of our work.
>
> We have thoroughly considered your feedback and carefully responded to each of your questions. We would greatly appreciate your feedback on whether our responses have addressed your concerns to your satisfaction.
>
> Once again, we sincerely thank you for your invaluable contribution to our paper. As the rebuttal phase is progressing, we eagerly await your post-rebuttal feedback.
>
> Best regards,
> Authors.

---

### Official Review · Reviewer_zBbT · 2023-07-10

**Soundness:** 2 fair
**Presentation:** 3 good
**Contribution:** 1 poor
**Rating:** 3
**Confidence:** 4

**Summary:**

The authors compared various deep learning and traditional non-deep machine learning model performance on multiple molecular property prediction tasks. Their findings indicate that deep models tend to underperform non-deep ones due to the irregular patterns of molecular data and relatively small data size. To that end, they proposed to apply a feature mapping technique to help deep models capture high frequency patterns, and demonstrated the effectiveness on benchmark datasets.

I find this work more like a technical report providing comprehensive benchmark results. My main concerns include (1) the conclusions are somewhat expected and probably already well-known in the community (e.g., the sharp vs smooth decision boundary issue); (2) it seems that the main focus of this work is about feature engineering, while the proposed methodology (i.e., Fourier-type feature mapping) exhibits insufficient novelty.

Therefore, I do not recommend publishing this manuscript to NeurIPS.

**Strengths:**

- The authors provided comprehensive benchmark results on many commonly-used molecular property prediction datasets.
- Comparing the performance amongst non-deep and deep learning models on molecular property prediction is a significant research topic.


**Weaknesses:**

- Insufficient novelty. The proposed methodology is well-known (e.g., NeRF, Fourier features, etc.).
- The main idea is about feature engineering, while the set of optimal features cannot be theoretically guaranteed.
- The comparison between deep and non-deep models on these molecular datasets might be unfair for deep models due to the data size. Although the authors claimed that irregular data patterns is another important factor explaining their predictive power, larger data size (i.e., the scaling law) indeed tends to improve deep model generalizability (e.g., see [pvd](https://arxiv.org/abs/2206.00133)).

**Questions:**

- The molecular similarity from a visual perspective (e.g., Figure 3) is quite intuitive for us, but does it necessarily mean deep models cannot find an appropriate latent space where these "cliff molecules" are far apart?
- With increasing data size, what is the trend of deep vs non-deep model performance?
- Since the conclusion is that tree-based models tend to outperform others, does it mean rule-based models are superior to deep models?
- I think Figure 2 and 4 are a bit misleading. The scale of y-axis has been engineered to enhance the numerical difference, imo this is not appropriate, especially for AUROC plots (i.e., baseline 0.5).
- Line 245, the authors mentioned "convex combination", yet my understanding is that $\hat{x}_i$ underwent orthogonal transformation Q from $x$, which is not necessarily a convex combination of $x$?

**Limitations:**

- The over-smoothing issue of GNNs could be discussed in the manuscript to further support their claims.
- For molecules without available 3D structures, RDKit-generated conformers are used. However, these generated structures (probably relaxed with MLFF) are not as accurate as DFT-relaxed ones -- which might confuse 3D structure-based deep models.

---

> ### Author Response · Authors · 2023-08-15
> **Response to Reviewer zBbT (1/3)**
>
> > *Insufficient novelty. The proposed methodology is well-known (e.g., NeRF, Fourier features, etc.).*
>
> Thanks for your helpful reviews! We have addressed the novelty of our proposed methodology, IFM, by highlighting the key differences between our method and other feature mapping techniques. In Line 274-275 and Section 9 of the appendix, we have provided a clear explanation. Specifically,
>
> In Nerf's feature embedding (Sinusoidal Feature Mapping), the feature mapping is deterministic and only includes on-axis frequencies. This inherent bias makes it more suitable for data that has a greater concentration of frequency content along the axes. In contrast, our IFM allows all directions to share the same frequency content, providing a more comprehensive and flexible representation. We have also provided theoretical justifications for the superiority of our IFM method over this technique in the appendix.
>
> In Gaussian Feature Mapping (Fourier features), the matrix-vector product can inadvertently mix different dimensions of molecular features, which is an undesirable characteristic. In contrast, our IFM is designed to avoid this issue. Besides, while the parameters $\mathbf{B}$ in Gaussian Feature Mapping are sampled from a fixed Gaussian distribution, the parameters $\mathbf{c}$ in IFM are learnable. This allows our method to adapt and optimize the feature mapping process based on the specific dataset.
>
> Thank you once again for your valuable feedback!
>
> > *The main idea is about feature engineering, while the set of optimal features cannot be theoretically guaranteed.*
>
> Thanks for your insightful reviews! Molecular property prediction tasks often involve multiple objectives, making it challenging to identify a single set of features that is optimal for all tasks. In our study, we have implemented a feature selection process based on the following criteria: (1) removal of features with missing values, (2) exclusion of features with extremely low variance (variance < 0.05), and (3) elimination of features exhibiting high correlation (Pearson correlation coefficient > 0.95) with another feature. These steps were taken to ensure the quality and relevance of the selected features for each dataset. Thank you for your valuable feedback!
>
> > *The comparison between deep and non-deep models on these molecular datasets might be unfair for deep models due to the data size. Although the authors claimed that irregular data patterns are another important factor explaining their predictive power, larger data size indeed tends to improve deep model generalizability.*
>
> **Table Re1**: The results (mean $\pm$ std) of representative models on OGB-molhiv (ROC-AUC score) and OGB-molpcba (the average precision score). The top-3 performance for each dataset is highlighted with **bold**.
> | Models| SVM |XGB|RF| CNN| RNN| TRSF| MLP| GCN| MPNN| GAT| D-MPNN| AFP | SPN|
> |----|----|----|----|----| ----  |---- |----|----|----| ----  |---- |---- |----|
> | OGB-molhiv (41,127) | 0.8253 $\pm$ 0.0036| **0.8297** $\pm$ 0.0051 | **0.8329** $\pm$ 0.0023 | 0.7915 $\pm$ 0.0083 | 0.7836 $\pm$ 0.0040| 0.8135 $\pm$ 0.0062| 0.8265 $\pm$ 0.0052| 0.8193 $\pm$ 0.0078| 0.8026 $\pm$ 0.0053| 0.8117 $\pm$ 0.0040|**0.8276** $\pm$ 0.0092| 0.8247 $\pm$ 0.0074|0.8018 $\pm$ 0.0067|
> | OGB-molpcba (437,929)| 0.2963 $\pm$ 0.0035| 0.3054 $\pm$ 0.0027 | **0.3072** $\pm$ 0.0019| 0.2547 $\pm$ 0.0014 | 0.2832 $\pm$ 0.0026 | 0.2759 $\pm$ 0.0033| **0.3058** $\pm$ 0.0026| 0.2811 $\pm$ 0.0023| 0.2785 $\pm$ 0.0019| 0.2816 $\pm$ 0.0037|0.3016 $\pm$ 0.0037| **0.3125** $\pm$ 0.0024|0.2955 $\pm$ 0.0038|
>
> Thanks for your helpful reviews! We evaluated 13 models on two larger molecule datasets OGB-molhiv (with 41,127 molecules) and OGB-molpcba (with 437,929 molecules) in Table Re1. As can be observed, non-deep models can also beat most deep models on the large molecule dataset (OGB-molpcba). Furthermore, we would like to highlight that the public leaderboard (available online) in OGB demonstrates that a combination of one molecule fingerprint and random forest can outperform many deep models. While it is true that some deep models achieve better performance on the leaderboard, they often rely on additional data, large-scale pre-training, or ensemble learning techniques.
>
> Also, in our paper (Line 178 - 180), we observe that all the non-deep models can outperform any deep ones on some larger-scale datasets (e.g., MUV and QM 7). However, in some small datasets (e.g., ClinTox and ESOL), some deep models can beat partial non-deep ones.
>
> Therefore, we argue that there is indeed another important factor contributing to the limitations of deep models, beyond the small size of molecule datasets. Moreover, it is worth noting that small-scale molecule datasets are more prevalent in practice, as labeling molecules typically relies on labor-intensive wet-lab experiments.
>
> Thank you for bringing up this point, and we appreciate your valuable feedback!

---

> > ### Author Response · Authors · 2023-08-15
> > **Response to Reviewer zBbT (2/3)**
> >
> > >*The molecular similarity from a visual perspective (e.g., Figure 3) is quite intuitive for us, but does it necessarily mean deep models cannot find an appropriate latent space where these "cliff molecules" are far apart?*
> >
> > Thank you for your comment! We would like to clarify that the purpose of Figure 3 was to provide an illustrative example of activity cliffs, where even small changes in the chemical structure of a molecule can result in significant variations in its bioactivity. In our experiments, we employed the Tanimoto coefficient of extended connectivity fingerprints [1] to calculate the structural similarity between molecules, which is a commonly used approach in computational chemistry.
> > Also, we want to emphasize that our intention was not to imply that deep models are unable to find an appropriate latent space where these “cliff molecules” are far apart. Rather, based on our experimental findings, we concluded that tree models are better suited for fitting non-smooth target functions mapping molecules to their bioactivity, such as those involving activity cliffs.
> >
> > > *With increasing data size, what is the trend of deep vs non-deep model performance?*
> >
> > Thanks for your helpful reviews! We have provided the analysis of the trend in Line 178-180. Specifically, all the non-deep models can outperform any deep ones on some larger-scale datasets (e.g., MUV and QM 7). However, in some smaller datasets (e.g., ClinTox and ESOL), some deep models can beat partial non-deep ones. Also, we evaluated 13 models on a larger molecule dataset OGB-molpcba (with 437,929 molecules) in Table Re1 (above). As can be observed, non-deep models can also beat most deep models on OGB-molpcba. Therefore, we conclude that there is indeed another important factor contributing to the limitations of deep models, beyond the small size of molecule datasets.
> >
> > >*Since the conclusion is that tree-based models tend to outperform others, does it mean rule-based models are superior to deep models?*
> >
> > Thanks for your contructive reviews! We do not mean that rule-based models are superior to deep models. Also, we want to clarify that tree models do not fall under the category of rule-based models. Rule-based models, as the name suggests, operate based on a set of predefined rules. These rules are typically expressed in the form of “if-then” statements, where specific conditions are checked, and corresponding actions or predictions are made. Examples of rule-based models include expert systems and rule-based classifiers. On the other hand, tree-based models use a hierarchical structure of decision nodes and leaf nodes to make predictions. They recursively split the input space based on feature values and create a tree-like structure. Each internal node represents a decision based on a specific feature, and each leaf node represents a prediction or outcome. Tree-based models learn the decision boundaries from the data and make predictions by traversing the tree structure.
> >
> > While tree models have shown better performance in handling non-smooth target functions, such as activity cliffs, compared to deep models in our study, it does not imply that rule-based models are superior to deep models in general.
> >
> >
> > >*I think Figure 2 and 4 are a bit misleading. The scale of y-axis has been engineered to enhance the numerical difference, imo this is not appropriate, especially for AUROC plots (i.e., baseline 0.5).*
> >
> > Thanks for your valuable feedback! we would like to clarify that the purpose of adjusting the y-axis scale was not to enhance the numerical difference artificially but rather to ensure better visualization of the performance differences between the methods being compared, especially considering that the performance metrics on different datasets may vary in scale.
> >
> > > *Line 245, the authors mentioned "convex combination", yet my understanding is that  underwent orthogonal transformation Q from, which is not necessarily a convex combination of ?*
> >
> > Thank you for your careful review and correction! You are absolutely correct, and we apologize for the error in our statement. It should indeed be a linear combination rather than a convex combination. The orthogonal transformation Q applied to the molecular features is a linear combination of the original features, which does not necessarily preserve convexity. Additionally, we want to assure you that this error does not affect our overall conclusion regarding the potential mixing of different dimensions of molecular features by deep models. We appreciate your valuable input, as it helps us improve the accuracy and clarity of our research. Thank you once again for your contribution.

---

> > > ### Author Response · Authors · 2023-08-15
> > > **Response to Reviewer zBbT (3/3)**
> > >
> > > >*The over-smoothing issue of GNNs could be discussed in the manuscript to further support their claims.*
> > >
> > > Thanks for your valuable feedback! Oversmoothing issue would degrade the performance of GNNs in **node-level task**. However, a recent work [2] theoretically validates that over-smoothing issues (deeper GNN layers) have little influence on the performance in **graph-level tasks** including the molecular property prediction here.
> > >
> > > > *For molecules without available 3D structures, RDKit-generated conformers are used. However, these generated structures (probably relaxed with MLFF) are not as accurate as DFT-relaxed ones -- which might confuse 3D structure-based deep models.*
> > >
> > > Thank you for your feedback! We understand your concern about using RDKit-generated conformers for molecules without available 3D structures. While these conformers may not be as accurate as DFT-relaxed structures, our results indicate that even for datasets (QM 7, QM8, and QM9) with DFT-relaxed structures, non-deep models without conformers can outperform 3D structure-based deep models.
> > >
> > > To address your concern and provide a comprehensive analysis, we can consider removing SphereNet results on RDKit-generated conformers. This will allow us to focus on comparing non-deep models and 3D structure-based deep models using DFT-relaxed conformers.
> > >
> > > We appreciate your valuable input, as it helps us improve our methodology and ensure the accuracy of our research. Thank you for your contribution!
> > >
> > > [1] Extended-Connectivity Fingerprints (JCIM 2020)
> > > [2] Search to Capture Long-range Dependency with Stacking GNNs for Graph Classification. (WWW 2023)
> > >
> > > ---
> > > We greatly appreciate your insightful comments, as they will undoubtedly help us improve the quality of our article. If our response has successfully addressed your concerns and clarified any ambiguities, we respectfully request that you consider raising the score of our article. Should you have any further questions or require additional clarification, we would be delighted to engage in further discussion. Once again, we sincerely appreciate your time and effort in reviewing our manuscript. Your feedback has been invaluable in improving our research.

---

> ### Author Response · Authors · 2023-08-18
> **Look forward to post-rebuttal feedback**
>
> Dear Reviewer zBbT,
>
> We would like to express our sincere gratitude for dedicating your time to reviewing our paper. Your insightful comments and suggestions have greatly contributed to enhancing the quality and clarity of our work.
>
> We have thoroughly considered your feedback and carefully responded to each of your questions with extensive experiments. We would greatly appreciate your feedback on whether our responses have addressed your concerns to your satisfaction.
>
> Once again, we sincerely thank you for your invaluable contribution to our paper. As the rebuttal phase is progressing, we eagerly await your post-rebuttal feedback.
>
> Best regards,
> Authors.

---

> > ### Comment · Reviewer_zBbT · 2023-08-20
> >
> > Dear authors,
> >
> > Thanks for your response. I went through the rebuttal and discussions with other reviewers and decided to maintain my scores. I think this work could be submitted to the dataset & benchmark track instead.
> >
> > Best

---

> > > ### Author Response · Authors · 2023-08-21
> > > **Thanks for your post-rebuttal feedback**
> > >
> > > Dear reviewer zBbT,
> > >
> > > Thanks for your valuable advice! It is worth noting that the datasets & benchmark track typically feature papers that primarily introduce new datasets without novel approaches. Given that our work introduces a method to improve the performance of deep models, we believe that the main track is the appropriate venue to showcase our research.
> > >
> > > Given the fact that you keep your rating as  ‘reject’, we want to know which specific concerns you still have (in addition to submission to the dataset & benchmark track). We remain dedicated to addressing them with the utmost eagerness.
> > >
> > > Thanks again for your valuable advice in terms of clarity and details (convex combination), especially the experiments on larger datasets.  Look forward to further discussions!
> > >
> > > Best,
> > > Authors.

---

### Author Response · Authors · 2023-08-15
**General response to all the reviewers**

Dear reviewers,

We would like to express our sincere gratitude for your valuable feedback on our manuscript. We apologize for the delay in responding to your comments as we needed additional time to conduct extensive experiments and address the issues raised.

We have carefully reviewed your suggestions and would provide detailed responses to each of your points. We appreciate your insights and the references you provided, which will greatly enhance the clarity and completeness of our manuscript.

If you have any further questions or concerns, please don’t hesitate to let us know. We are actively engaged in the discussion and are committed to improving the quality of our work. Thank you once again for your valuable input!

Best regards,
Authors

---

### Author Response · Authors · 2023-08-17
**Look forward to post-rebuttal feedback**

Dear reviewers,

Thank you for your insightful comments and feedback on our manuscript. We greatly appreciate the time and effort you have dedicated to reviewing our work. We have carefully  responded to your questions/concerns and have conducted extensive experiments to address them.

As the discussion deadline is approaching, we want to check if our response have adequately address your concerns. Also, if you have any further questions or concerns, we remain dedicated to addressing them with the utmost eagerness.

Once again, we sincerely appreciate your valuable input, and we look forward to hearing from you soon.

Best regards,
Authors.

---

### Decision · Program_Chairs · 2023-09-21

**Decision:**

Accept (poster)

**Comment:**

This paper studies the phenomenon that deep models are not as good as non-deep ML methods on molecule property prediction tasks.

The paper present empirical evidence for this, although this phenomenon is known before. The authors give plausible explanation: (1) non-smoothness nature of molecules; (2) better decision needs to be done for individual molecule features before mix.

The paper then proposes a method to improve the performance of all existing deep models (Independent Feature Mapping). This is a small extension of Tancik et al, but Tancik was never used for molecule tasks.

There are divided reviews for this paper. So making a decision is extremely difficult in this case. It can certainly go either way. Both sides have valid points which I would summarize below:

**Reason to accept**

S1. Understanding the behavior of deep models comparing to non-deep models for molecule property prediction tasks is very important and timely.

S2. The author’s contributions are therefore important since they provide a plausible explanation backed by numerous well-designed demonstrations as well as a proposed method based on the new understanding, again backed by convincing experiments.

S3. Additional experiments provided in the response are adequate to address major concerns of reviewers.

**Reason to Reject**

W1. It is not fully confirmed that in general case non-smoothness of data cause the issue of deep models comparing to non-deep models.

W2. The phenomenon is well known. Therefore the paper over-claim on this.

W3. The method is a straightforward extension to Tancik et al, and it is just feature engineering, therefore it lacks novelty.

W4. It is better suited for benchmark and data track.

W5. The authors did not provide official rebuttal in time, instead they provided comments during discussion period therefore reducing time to discuss.

W6. Too much significant results in author response so it will change the paper dramatically

I spend additional time to go through the paper, all the reviews, author responses and discussion. I agree with reviews on the above S1-3, W1 and W2. But W3-W5 are invalid or unsupported.
- Regarding W1. The paper does not provide a explanation why non-smoothness of data causes trouble for general deep models versus non-deep models. But it might be a too broad request for authors and the author did not claim that for general tasks as well. They only study molecule tasks.

- Regarding W3, the paper's IMF method is still different from Tancik et al, as explained by authors. The Tancik et al was for graphics/cv but not tested on Molecules. It is a good contribution even to demonstrate the effectiveness of Tancik et al on molecule tasks and this paper goes beyond that.

- W4 (provided by Reviewer rZmN, Reviewer ah6B) is invalid. This paper provides explanation of deep models and a new method to improve. It does not provide a benchmark or an evaluation metric. Reviewer rZmN, ah6B either misunderstood the main content (but from the review it seems they understand well) or misunderstood the Benchmark track.

- W5. The authors' response came late. It is not ideal and should be discouraged in the future. But we managed to have intensive discussion among reviewers.

- W6. It seems ok since these experiments are requested by reviewers. It simply does not make sense to reject a paper because author responded with the information requested by reviewers. It could be better if the original paper already included all these results.

Overall, the strengths outweigh the weakness. I support inclusion of the paper in the conference. It would be beneficial for the researchers studying this topic.

Suggestion for authors:
1. You may want to add the additional explanation and experimental results in the revision.
2. Please add more details about IMF method. You may shrink a bit in Section 1, 2, 3.
3. Please tune down your claims and avoid exaggeration  (the findings in section 3, which was discovered before. Non-smoothness explanation, please restrict scope, novelty of IMF and relation with Tancik et al.)